# Multi-pathway DNA-repair reporters reveal competition between end-joining, single-strand annealing and homologous recombination at Cas9-induced DNA double-strand breaks

Bert van de Kooij[1,2], Alex Kruswick[2], Haico van Attikum [1]✉ & Michael B. Yaffe [2,3,4]✉

DNA double-strand breaks (DSB) are repaired by multiple distinct pathways, with outcomes ranging from error-free repair to mutagenesis and genomic loss. DSB-repair pathway cross-talk and compensation is incompletely understood, despite its importance for genomic stability, oncogenesis, and genome editing using CRISPR/Cas9. To address this, we constructed and validated three fluorescent Cas9-based reporters, named DSB-Spectrum, that simultaneously quantify the contribution of multiple DNA repair pathways at a DSB. DSB-Spectrum reporters distinguish between DSB-repair by error-free canonical non-homologous end-joining (c-NHEJ) versus homologous recombination (HR; reporter 1), mutagenic repair versus HR (reporter 2), and mutagenic end-joining versus single strand annealing (SSA) versus HR (reporter 3). Using these reporters, we show that inhibiting the c-NHEJ factor DNA-PKcs increases repair by HR, but also substantially increases mutagenic SSA. Our data indicate that SSA-mediated DSB-repair also occurs at endogenous genomic loci, driven by Alu elements or homologous gene regions. Finally, we demonstrate that long-range end-resection factors DNA2 and Exo1 promote SSA and reduce HR, when both pathways compete for the same substrate. These new Cas9-based DSB-Spectrum reporters facilitate the comprehensive analysis of repair pathway crosstalk and DSB-repair outcome.

Double-strand DNA breaks (DSBs) are severe genotoxic lesions that need to be correctly repaired to prevent mutations, loss of genomic information, or devastating chromosomal rearrangements. DSBs result from exogenous sources like environmental radiation or anticancer chemotherapeutics, or from endogenous sources like replication stress or endogenous nucleases that function during meiosis and immune cell maturation[1]. Furthermore, site-specific DSB-formation by Cas9 or related endonucleases is central to CRISPR-based genome editing, which has become an essential technology in biomedical research, and holds great promise to cure disease in gene therapy applications[2–4].

Mammalian cells are equipped with multiple pathways to repair DSBs[5]. In most cells, the dominant DSB-repair pathway is canonical Non-Homologous End-Joining (c-NHEJ)[5]. DSB-repair by c-NHEJ is

mostly accurate, but can also introduce small insertions and deletions (InDels) at the DSB junction caused by minor editing of the DSB ends prior to ligation[6]. A second DSB-repair pathway is Homologous Recombination (HR), which plays a particularly important role during the S/G2 phases of the cell cycle. HR starts with the resection of the DSB ends to generate single-strand 3' overhangs that subsequently invade homologous DNA, usually the sister chromatid, which functions as a template that is copied in the downstream repair steps[7]. HR is therefore considered an error-free pathway. However, DNA fragments that share homology with the DSB-site but carry mutations or even large insertions can also be copied during repair by HR. This is utilized for genome editing purposes by co-delivery of a repair template with the desired mutant sequence. Compared to genome editing by c-NHEJ, this expands the editing possibilities and allows additional control over the editing outcome, but this approach is hampered by the low frequency of HR compared to c-NHEJ[5]. Many strategies to increase the ratio of HR to c-NHEJ have been developed, but c-NHEJ is surprisingly robust and remains the dominant repair pathway even when applying these HR-promoting methods[8].

In addition to c-NHEJ and HR, DSBs can be repaired by alternative End-Joining (a-EJ), or Single-Strand Annealing (SSA)[9,10]. Both pathways require end-resection to expose regions of homology on the opposing ends of the broken DNA molecule. Annealing of these homologous regions is followed by removal of non-homologous resected DNA, polymerase-mediated fill-in, and ligation. Compared to a-EJ, SSA requires larger regions of homology and is more tolerant to large stretches of DNA separating the homologous regions[11]. Both a-EJ and SSA are inherently mutagenic, and SSA, in particular, can result in the loss of multiple kilobases of genetic information[12]. How frequently these pathways act on genomic DSBs is not known, but recent studies suggested that a substantial fraction of Cas9-generated DSBs may be repaired by a-EJ[13,14]. The contribution of SSA to the repair of Cas9 DSBs, however, has not been thoroughly addressed.

These mechanistically distinct DSB-repair pathways do not function independently, but act in a network that is tightly regulated to optimally preserve the integrity of the genome. Deregulation of the DSB-repair network reduces genomic stability and can have severe pathogenic consequences, most notably cancer development[15]. Intense research efforts are slowly revealing the organization of this network and have shown that end-resection dependent repair by HR is promoted by BRCA1, whereas the 53BP1-Shieldin complex inhibits end-resection and therefore directs DSB-repair towards c-NHEJ[16–21]. Nevertheless, many nodes and edges of the DSB-repair network remain to be revealed. For example, whereas it has generally been well described how inhibiting c-NHEJ, or promoting resection of DSB ends, affects repair by HR, most studies have not reported how this affects repair by other end-resection dependent pathways like a-EJ, or SSA (reviewed in ref. [8]). Moreover, how DSB-repair pathway choice between a-EJ, SSA, and HR is regulated after the initiation of end-resection is not well described. These are important questions to answer since there is a strong interest in controlled manipulation of DSB-repair pathway choice to improve anticancer therapy and, more recently, to direct genome editing outcome[8,22].

To facilitate network-level analysis of DSB-repair activity, we developed three variants of a novel genomic Cas9-based DSB-repair reporter construct that we named DSB-Spectrum. Reporter constructs that quantify the activity of DSB-repair pathways, such as DR-GFP, which quantifies the frequency of HR, have greatly contributed to DSB-repair research[23]. Multiple elegant variations of DR-GFP-type reporters have been generated that quantify repair by c-NHEJ, a-EJ, or SSA[24–26], and in some cases, a combination of these pathways[27]. Using these published reporter systems as a basis, we generated the DSB-Spectrum variants, which are multi-pathway reporters that allow for simultaneous quantification of the frequency of DSB-repair by end-joining, SSA, and HR.

These DSB-Spectrum reporters display high frequencies of DSB-repair through all pathways, function as single constructs without the requirement for ectopic HR-donors, and are activated by Cas9-induced DSB generation, allowing for direct translation of the results to CRISPR-strategies. DSB-Spectrum efficiently and correctly reveals DSB-repair pathway crosstalk between NHEJ, SSA, and HR. We show that SSA can strongly contribute to DSB-repair and is significantly promoted by inhibition of DNA-PKcs. Our data indicate that repetitive elements in the human genome can drive SSA of Cas9-induced DSBs in endogenous genomic loci. Finally, we demonstrate that SSA, but not necessarily HR, is promoted by the long-range end-resection factors Exo1 and DNA2, thus providing a potential mechanism to direct DSB-repair away from SSA towards HR. Our studies show that DSB-Spectrum captures the complexity of the DSB-repair network and detects DSB-repair phenotypes that can easily be missed by studies that focus on individual repair pathways.

## Results

### DSB-Spectrum_V1 is a multi-pathway DSB-repair reporter that simultaneously quantifies c-NHEJ and HR

To study the interaction between the various DSB-repair pathways, we sought to design a multi-pathway reporter in which repair of a single site-specific DSB by each individual pathway would result in a unique, pathway-specific expression pattern of fluorescent proteins (Fig. 1a). Such a reporter would reveal, on an individual cell basis, which pathway was used to repair the DSB. Within a multicellular population, such a reporter system could be used to simultaneously quantify the frequency of repair by multiple DSB-repair pathways using flow cytometry as a readout.

To accomplish this, we first modified and combined two existing reporter systems to create a multi-pathway reporter construct designed to detect distinct repair products created by either c-NHEJ or HR (Fig. 1a).[23,25] Fig. 1b shows this reporter, which we named DSB-Spectrum_V1, and which consists of a CMV promoter followed by a modified gene encoding Blue Fluorescent Protein (BFP), a 2.6 kb intervening region, and a truncated gene encoding part of Green Fluorescent Protein (GFP) and lacking a promoter. Hence, no GFP is expressed by the reporter. No functional BFP is produced either because the BFP gene contains a 46 bp spacer insert separating the gene at the triplet encoding for Gly-68, immediately adjacent to the critical amino acid, His-67, responsible for blue fluorescence. This spacer sequence can be excised by targeting Cas9 to the edges using sequence-specific guide RNAs, leaving behind a blunt-ended DSB. Ligation of the DSB through error-free c-NHEJ would restore the intact BFP sequence and result in BFP expression. Alternatively, the DSB can be repaired by Homologous Recombination (HR) using the truncated GFP gene as a repair template since it shares high sequence homology with the BFP gene (Supplementary Fig. 1a, b). Repair by HR will thus result in gene conversion and expression of GFP (Fig. 1b). Error-free c-NHEJ and HR can therefore be clearly distinguished from mutagenic repair pathways because these latter types of repair would not result in BFP or GFP expression (Fig. 1a).

To obtain a homogeneous reporter cell-line carrying a single copy of the reporter integrated into the genome, HEK 293T cells were infected with DSB-Spectrum_V1 containing lentivirus at low multiplicity of infection, followed by expansion of a single-cell clone. For this DSB-Spectrum_V1 clone and for other DSB-Spectrum clones used in this manuscript, Splinkerette PCR was used to map the genomic integration site of the reporter (Supplementary Fig. 2a)[28]. For all clones, the sites of integration of the reporter construct were within large introns of genes not associated with DNA-repair (Supplementary Fig. 2a). To validate the mapped integration sites, genomic DNA was isolated from the DSB-Spectrum cell-lines, or the parental cell-line as a control, followed by restriction enzyme-mediated digestion and Southern blotting (Supplementary Fig. 2b). A single product was

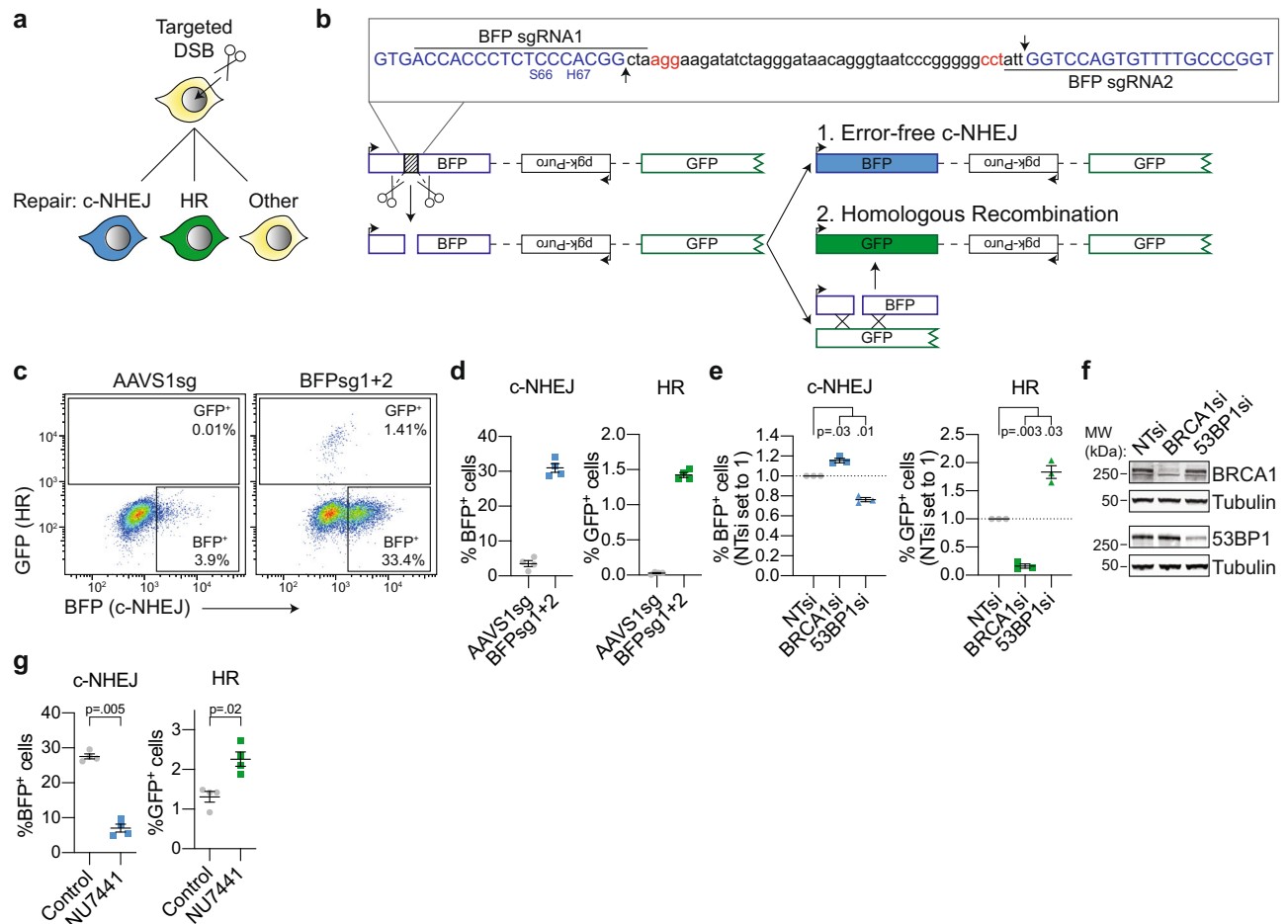

**Fig. 1 | DSB-Spectrum_V1 is a reporter for both c-NHEJ and HR. a** Cartoon depicting potential outcomes of a multi-pathway reporter cell-line designed to quantify DSB-repair by error-free c-NHEJ and HR. **b** Diagramatic representation of the genomic DSB-repair reporter construct DSB-Spectrum_V1. Expanded region shows the DNA sequence targeted by Cas9. The sequence of the BFP cDNA is displayed in blue, the PAM sequences of the sgRNA target sites are displayed in red. Arrows in the inset and scissors in the cartoon indicate the Cas9 cut sites. Ligation of the distal DSB ends by error-free c-NHEJ will remove the spacer sequence and restore the BFP gene. **c, d** DSB-Spectrum_V1 was integrated into the genome of HEK 293T cells by lentiviral infection, followed by expansion of a single-cell clone. The resulting DSB-Spectrum_V1 cell-line was transfected with Cas9 and an sgRNA targeting a control locus (AAVS1) or BFP, and analyzed by flow cytometry at 72 h after transfection. Panel **c** shows representative flow plots, and panel **d** shows the

quantification of multiple experiments ($n = 4$; mean ± SEM). **e** DSB-Spectrum_V1 cells were transfected with indicated siRNAs (NTsi = Non-Targeting control), followed by transfection with Cas9 and an sgRNA targeting a control locus or BFP. At 72 h after Cas9 transfection cells were analyzed by flow cytometry. Percentages of each fluorescent population in the BFPsg-transfected cells were corrected for the background percentages seen in the AAVS1sg-transfected cells. Depicted is the ratio of background-corrected percentages for each fluorescent population to that of the NTsi control ($n = 3$; mean ± SEM; One-way ANOVA, post-hoc Dunnett's). **f** Western blot of lysates from cells analyzed in panel **e**. Tubulin is used as loading control. **g** As in panel **d**, but including treatment with NU7441 (2 μM), and background-corrected as described in panel **e** ($n = 4$; mean ± SEM; ratio paired $t$-test, two-tailed). Source data for panels **d, e,** and **g** are provided as a Source Data file.

detected by two different southern probes binding the DSB-Spectrum cassette using two different digestion strategies (Supplementary Fig. 2b). Furthermore, the migration pattern of the detected products matched the product size calculated based on the mapped integration sites (Supplementary Fig. 2a, b). This demonstrated the integration of a single copy of the full-length reporter in all clones.

Next, the HEK 293T DSB-Spectrum_V1 cells were transfected with Cas9 cDNA, and either an sgRNA targeting a control genomic locus (AAVS1sg) or two sgRNAs targeting the spacer region in the reporter (BFPsg1 + 2), followed by flow cytometric analysis 72 h post-transfection. The Cas9-sgRNA construct also contained a fluorescent protein-encoding gene, either mCherry or iRFP(670), to monitor transfection efficiency and allow for gating on transfected cells (Supplementary Fig. 3a). As shown in Fig. 1c, d, distinct BFP+ and GFP+ populations could be detected specifically upon targeting Cas9 to the reporter. Approximately 30% of cells were BFP+, while ~1.5% of cells were GFP+, consistent with a minority of DSBs being repaired by HR

compared to error-free c-NHEJ (Fig. 1d). To validate that the BFP+ and GFP+ populations resulted from repair of the Cas9 DSBs by c-NHEJ or HR, respectively, we used RNAi to deplete either BRCA1, an HR-promoting factor, or 53BP1, which is a well-established c-NHEJ-promoting factor. Depletion of BRCA1 strongly reduced the frequency of GFP+ cells, while it increased the frequency of BFP+ cells (Fig. 1e, f; see Source Data file for uncropped blots). The opposite phenotype was observed upon loss of 53BP1 (Fig. 1e, f). These data are fully consistent with the known functions of 53BP1 and BRCA1 in DSB-repair[5] and confirm that the BFP+ and GFP+ populations resulted from end-joining and HR, respectively.

To further validate DSB-Spectrum_V1 as a reporter for c-NHEJ, rather than other types of end-joining, we individually silenced the expression of the core c-NHEJ factors Ku80, DNA-PKcs, XRCC4, or Lig4, or silenced expression of the a-EJ factor Polθ. Depletion of each of the c-NHEJ factors significantly reduced the frequency of BFP+ cells, while no such effect was observed after depletion of Polθ, consistent

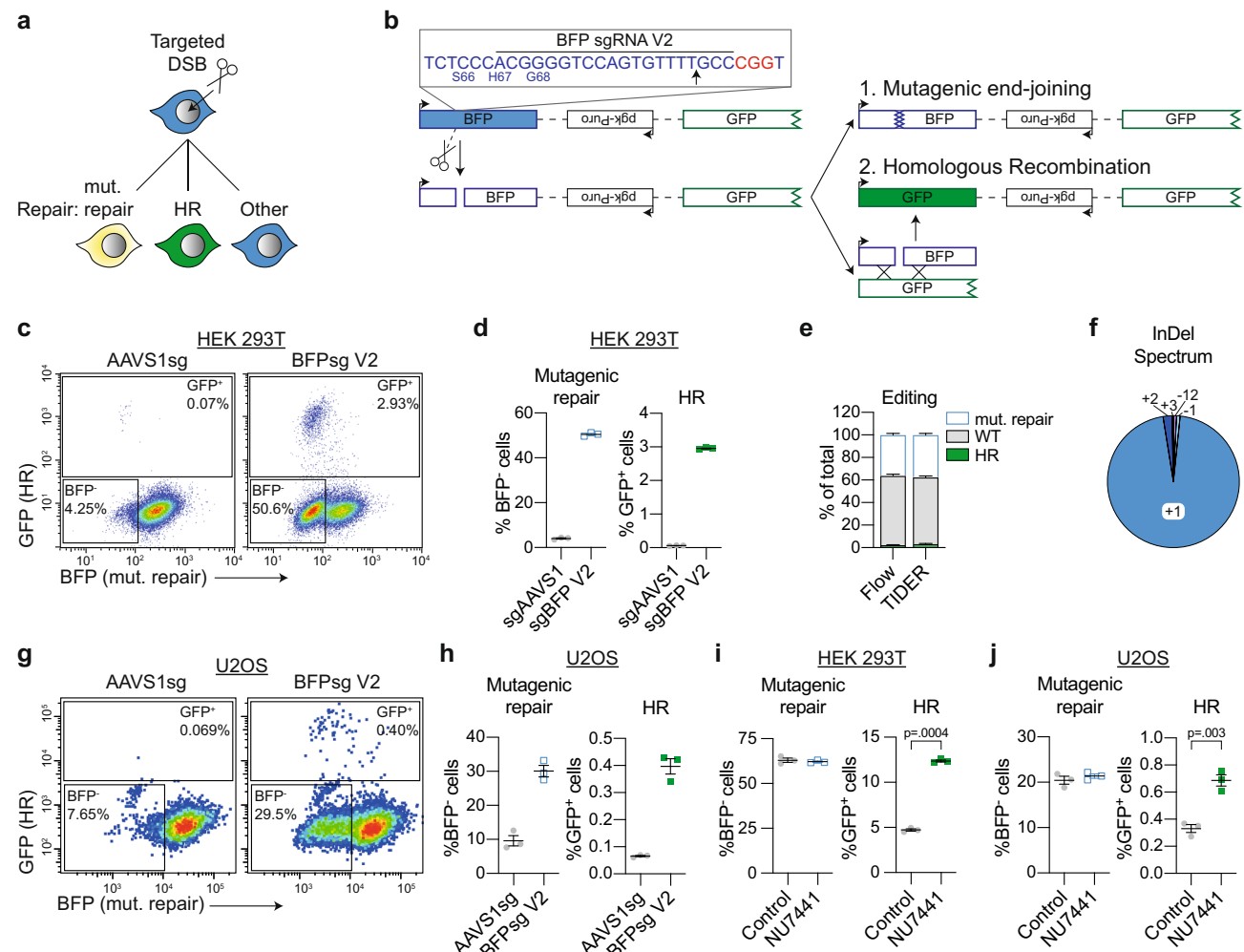

**Fig. 2 | DSB-Spectrum_V2 is a reporter for both mutagenic repair and HR. a** Cartoon depicting potential outcomes of a multi-pathway reporter cell-line designed to quantify DSB-repair by mutagenic end-joining and HR. **b** Diagramatic representation of the genomic DSB-repair reporter construct DSB-Spectrum_V2. Expanded region shows the DNA sequence targeted by Cas9. The sequence of the BFP cDNA is displayed in blue, the PAM sequence of the sgRNA target site is displayed in red. Arrow in the inset and scissors in the cartoon indicate the Cas9 cut site. **c, d** Flow cytometry plots and quantification of mutagenic repair and HR in DSB-Spectrum_V2 cells (*n* = 3; mean ± SEM), as in Fig. 1c, d. **e** DSB-Spectrum_V2 cells were transfected with Cas9 and an sgRNA targeting either a control locus or BFP. At 48 h after Cas9 transfection genome

editing was quantified by TIDER analysis of the sequenced target site. At 72 h after Cas9 transfection cells were analyzed by flow cytometry (*n* = 3; mean ± SEM). **f** DSB-Spectrum_V2 cells were transfected with Cas9 and an sgRNA targeting BFP, and 72 h later the BFP⁻ population was collected by FACS. InDel frequency was determined by TIDER analysis of the sequenced target site. **g, h** As in panels **c** and **d**, but for U2OS DSB-Spectrum_V2 cells (*n* = 3; mean ± SEM). **i** Mutagenic repair and HR was quantified in HEK 293T DSB-Spectrum_V2 cells with or without treatment with NU7441 (2 μM; *n* = 3; mean ± SEM; ratio paired *t*-test, two-tailed). **j** As in panel **i**, but for U2OS DSB-Spectrum_V2 cells (*n* = 3; mean ± SEM; ratio paired *t*-test, two-tailed). Source data for panels **d, e, f, h, i**, and **j** are provided as a Source Data file.

with BFP expression resulting from DSB-repair by c-NHEJ rather than by a-EJ (Supplementary Fig. 3b-d). Next, we assessed the reporter phenotype following treatment with NU7441, a small molecule inhibitor of DNA-PKcs kinase activity[29]. NU7441 treatment resulted in a more than three-fold reduction in the frequency of error-free repair by c-NHEJ (Fig. 1g). It also resulted in a close to a two-fold increase in the percentage of GFP⁺ cells, demonstrating that loss of c-NHEJ is compensated for by an increase in HR. Taken together, these results show that DSB-Spectrum_V1 can reveal the interdependence and crosstalk between the c-NHEJ and HR DSB-repair pathways within a single population of cells.

## DSB-Spectrum_V2 quantitatively reports mutagenic end-joining and HR

To generate a complementary reporter system capable of quantifying HR together with mutagenic end-joining, rather than error-free c-NHEJ, a variant of DSB-Spectrum_V1 was developed (Fig. 2a, b). The resulting

DSB-Spectrum_V2 expresses a functional BFP gene under basal conditions, in contrast to the spacer-separated variant encoding non-functional BFP in DSB-Spectrum_V1. A single sgRNA can then be used to target Cas9 to the sequence in the BFP gene that differs from GFP and is essential for its blue fluorescence (Supplementary Fig. 1a, b). Following generation of a DSB by Cas9, repair by mutagenic end-joining, which could be either mutagenic c-NHEJ or a-EJ, can be monitored by loss of BFP expression while HR repair, as in DSB-Spectrum_V1, can be monitored as a gain of GFP expression. Repair of the DSB by error-free c-NHEJ restores BFP expression, but this repair mechanism cannot be quantified with this reporter construct since a similar phenotype would result from failure of Cas9 cleavage (labeled 'other' in Fig. 2a).

Following the integration of a single copy of DSB-Spectrum_V2 into the genome of HEK 293T cells, a clonal cell-line was generated (Supplementary Fig. 2a, b). Cells were then transfected with Cas9 together with an sgRNA targeting either a control genomic locus

(AAVS1sg), or BFP, and DSB-repair outcomes were monitored by flow cytometric analysis. As shown in Fig. 2c and d, DSB-Spectrum_V2-expressing cells transfected with the control sgRNA were BFP⁺, consistent with the design of the reporter construct. Targeting of Cas9 to the reporter resulted in a substantial fraction of the cells losing BFP expression (±50%), while a minor fraction switched from expressing BFP to expressing GFP (almost 3%). Loss of BFP fluorescence requires degradation and cell division-mediated dilution of the pre-existing BFP protein pool present at the time of mutagenic editing of the *BFP* gene. To determine the readout window that allows detection of BFP loss, we analyzed DSB-Spectrum_V2 cells at different time points after DSB induction (Supplementary Fig. 4a). This demonstrated that a BFP⁻ population can be detected at 48 h after Cas9 transfection, but that optimal separation between the BFP-positive and -negative populations is achieved from 72 h onwards after Cas9 transfection (Supplementary Fig. 4a).

To confirm that loss of BFP expression was the direct result of repair by mutagenic end-joining of the DSB, rather than other epigenetic or genetic mechanisms like promoter silencing or (partial) loss of the DSB-Spectrum reporter, the DNA surrounding the Cas9-cleavage site in DSB-Spectrum_V2 was PCR amplified, followed by Sanger sequencing of the pool of PCR products. The editing efficiency, as well as the InDel spectrum, was then determined by deconvolving the mixture of sequencing chromatograms by TIDER analysis[30,31]. The frequencies of wild-type (WT) BFP, mutated BFP, and GFP sequences detected by TIDER analysis reflected the respective frequencies of BFP⁺, BFP⁻ and GFP⁺ cells detected by flow cytometry (Fig. 2e). Hence, these direct sequencing results verify that loss of BFP was primarily caused by mutagenic end-joining at the DSB-site. TIDER analysis and sequencing of multiple individual target loci further revealed that the majority of edited repair products contain a +1 thymine insertion at the DSB junction (Fig. 2f; Supplementary Fig. 4b), which is typical of a 1 bp staggered overhang generated by spCas9 cutting, followed by c-NHEJ-dependent repair[13,14,32,33].

DSB-repair by c-NHEJ can result in gene mutations that fail to disrupt expression of a functional protein product, like silent mutations or in-frame InDels. To validate that none of the BFP⁺ cells contained such phenotypically silent mutagenic end-joining products, we performed a Surveyor nuclease assay[34]. The Cas9-target site was PCR amplified from the BFP⁺ as well as the BFP⁻ cells and each PCR product was denatured and re-annealed to itself. If the PCR product was obtained from a mixed pool of wild-type and mutant sequences, the annealing procedure will result in mismatched duplexes that can be digested by the Surveyor nuclease. As expected, Surveyor nuclease digestion products were detected in PCR products from the BFP⁻ population, while no such products were detected in the BFP⁺ population, indicating that these were all WT sequences (Supplementary Fig. 4c). Thus, the flow cytometry, sequencing, and Surveyor nuclease data demonstrate that DSB-Spectrum_V2 is an accurate reporter for both mutagenic end-joining and HR.

To expand the utility of the reporter system, we analyzed and validated the performance of DSB-Spectrum_V2 in another widely used cell-line and generated DSB-Spectrum_V2 U2OS cells (Supplementary Fig. 2a, b). As observed with the HEK 293T cell-line, both BFP⁻ and GFP⁺ populations were detected in this U2OS clonal cell-line specifically after targeting Cas9 to the reporter (Fig. 2g, h). Notably, the gene editing frequencies, in particular the frequency of HR, were lower in U2OS than in HEK 293T cells (Fig. 2d, h), and these editing frequencies were comparable between multiple independent U2OS DSB-Spectrum_V2 clones (Supplementary Fig. 4d). This suggests that the low editing frequencies are not dependent on the reporter integration site, but are a characteristic of U2OS cells. Nonetheless, as all repair populations were present and measurable, these data demonstrate that the DSB-Spectrum_V2 construct consistently functions as a DSB-repair reporter in multiple cell types.

The results obtained with DSB-Spectrum_V1 demonstrated that inhibition of DNA-PKcs with the small molecule inhibitor NU7441 reduced c-NHEJ and promoted HR (Fig. 1g). This experiment was repeated with HEK 293T DSB-Spectrum_V2 cells. As expected, a strong increase in HR was observed upon DNA-PKcs inhibition, but surprisingly, this did not result in the expected reduction in BFP loss (Fig. 2i, Supplementary Fig. 4e). Similar results were obtained in U2OS DSB-Spectrum_V2 cells, where treatment with NU7441 increased HR but failed to reduce mutagenic end-joining (Fig. 2j). These results suggest that, in DSB-Spectrum_V2 cells, the loss of c-NHEJ following NU7441 treatment is compensated for by an alternative type of mutagenic repair that results in BFP loss.

**DNA-PKcs inhibition promotes DSB-repair by SSA**

To explore the remaining types of mutagenic repair that might compensate for the loss of c-NHEJ in the presence of the DNA-PKcs inhibitor, we investigated whether there might be an increase in repair by a-EJ or SSA. To assess the contribution of a-EJ, an ~1100 base-pairs (bp) region surrounding the Cas9-cleavage site in DSB-Spectrum_V2 was PCR amplified from both control and NU7441-treated cells, sequenced, and subjected to TIDER analysis. This revealed that NU7441 treatment almost completely inhibited the generation of the dominant repair product containing a 1 bp insertion at the break junction (Fig. 3a, Supplementary Fig. 5a, also see Fig. 2f), further validating that the majority of mutagenic end-joining of the DSB-Spectrum_V2 target site is dependent on c-NHEJ, rather than on a-EJ. Two other c-NHEJ repair products were also detected, albeit with low frequency, which contained either a 1 bp deletion or a 2 bp insertion (Fig. 3a, Supplementary Fig. 5a). Furthermore, TIDER analysis identified repair products containing 6, 9, and 12 bp deletions, which were all increased in frequency in the NU7441-treated cells (Fig. 3a, Supplementary Fig. 5a). To test whether these products were generated by a-EJ, we silenced expression of the a-EJ factor PolΘ (PolQ; Supplementary Fig. 5b). This decreased the frequency of the 9 bp and 12 bp deletion products in NU7441-treated cells, indicating that they were generated by PolΘ-dependent a-EJ (Fig. 3a, Supplementary Fig. 5a). PolΘ knockdown did not affect the frequency of the 6 bp deletion product, which therefore must have been the result of a repair process that is neither c-NHEJ nor PolΘ-dependent a-EJ. Notably, the combined frequency of the 6 bp deletion, a-EJ, and remaining c-NHEJ repair products together was ~10%, which is considerably less than the ~70% of BFP-negative cells detected in the NU7441-treated cells by flow cytometry (Fig. 3b, left panel). In conclusion, this sequence analysis indicates that most of the loss of BFP expression observed following inhibition of mutagenic repair by c-NHEJ is not caused by an upregulation of a-EJ.

We next considered repair by SSA. The reporter contains two regions of homology, i.e., the 5′ coding regions of the BFP and GFP genes, that have the potential to anneal together during SSA repair, which would result in the removal of the about 3 kb intervening sequence separating the regions, including the pgk-Puro gene (Fig. 3c). The annealed product would generate cells that are both BFP⁻ and GFP⁻ since the resulting BFP/GFP-gene hybrid contains the C-terminally truncated region from the GFP gene (Fig. 3c). Repair by SSA would also remove the binding site for one of the primers used to PCR-amplify the Cas9 cut site for TIDER analysis (Fig. 3c, blue primer). Thus, these SSA repair products, if present, would not have been included in the TIDER analysis, which could explain the gross underestimation of the total BFP mutation frequency by this technique (Fig. 3b, left panel). Concomitantly, this would result in an overestimation of all other non-SSA repair products that were included in the TIDER analysis. Consistent with this reasoning, in the NU7441-treated cells, the frequency of WT and HR (GFP) sequences detected by TIDER analysis was substantially higher than the frequency of BFP⁺ and GFP⁺ cells detected by flow cytometry (Fig. 3b, middle and right panels, Supplementary Fig. 5a). Therefore, these

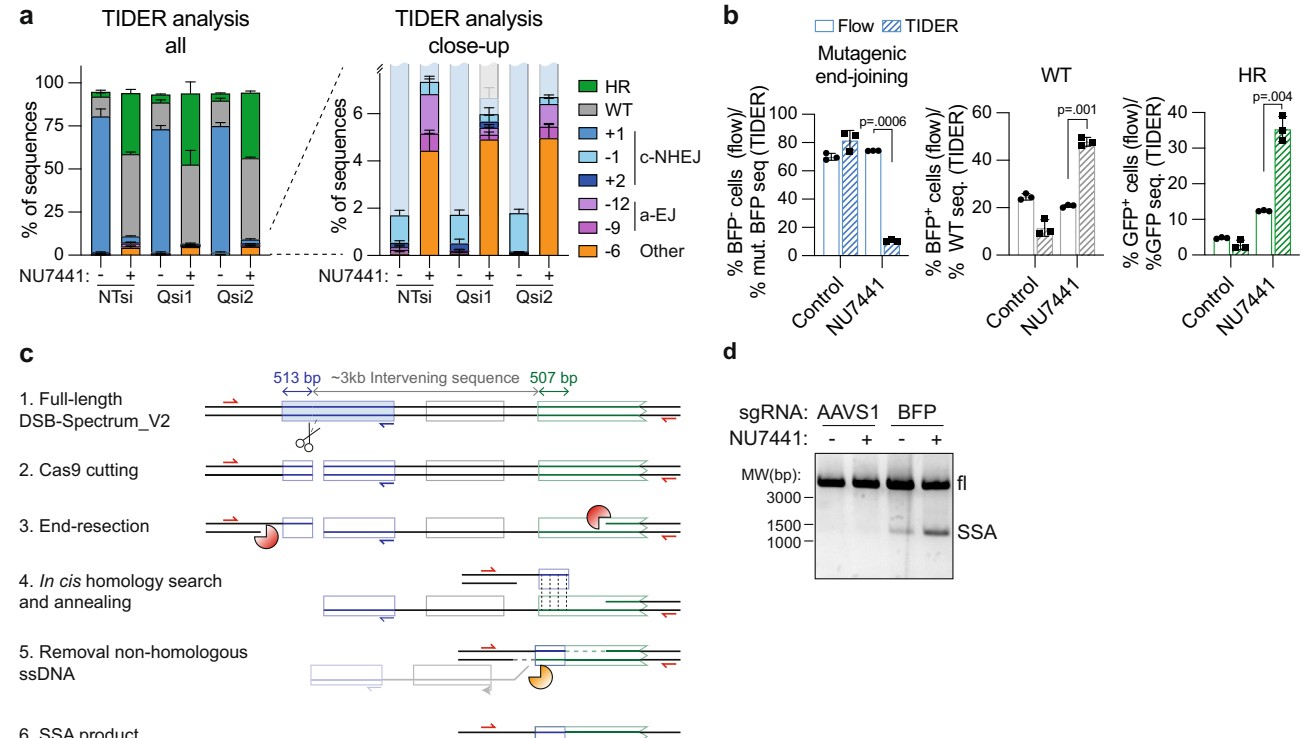

**Fig. 3 | Inhibition of DNA-PKcs promotes DSB-repair by single-strand annealing. a** HEK 293T DSB-Spectrum_V2 cells were transfected with indicated siRNAs (NTsi = Non-Targeting control, Qsi = PolQ-targeting siRNA), followed by transfection with Cas9 and an BFP-targeting sgRNA. Next, cells were treated with NU7441 (2 μM) or left untreated. At 48 h after transfection, Cas9-expressing cells were collected by FACS and subjected to sequence analysis of the target site using the TIDER algorithm. Frequency of all detected sequences is plotted in the left bar graph, a close-up of the bottom fraction of this graph is shown on the right (*n* = 3; mean ± SEM). **b** The data shown in panel **a**, and the data shown in Fig. 2i, were replotted to compare DSB-Spectrum_V2 editing frequencies in NU7441-treated cells as determined by either flow cytometry or TIDER-mediated sequence analysis (*n* = 3; mean ± SEM; ratio paired *t*-test, two-tailed). **c** Schematic

representation of repair of the Cas9-induced DSB in DSB-Spectrum_V2 by SSA between the 5′-end of the BFP gene (513 bp), located upstream of the DSB, and highly homologous 5′-end of the GFP gene (507 bp), located downstream of the DSB. Indicated are the primers (in red) to amplify the SSA repair product, the reverse primer (in blue) to amplify the target site for TIDER analysis, and the nucleases that perform end-resection (orange pacman) or ssDNA flap removal after annealing (yellow pacman). **d** DSB-Spectrum_V2 cells were transfected with Cas9 and an sgRNA targeting either the AAVS1 locus or BFP, and subsequently treated with NU7441 (2 μM). The genomic DNA was PCR amplified using the primers in red in panel **c**, and analyzed by DNA gel electrophoresis. A representative image of multiple independent experiments is shown. Source data for panels **a** and **b** are provided as a Source Data file.

results, taken together, suggest an increase in the presence of SSA repair products following DNA-PKcs inhibition.

To directly confirm repair by SSA, we designed a PCR strategy to amplify across the entire DSB-Spectrum locus (Fig. 3c, red primers). Using this strategy, amplification of a WT locus or a locus repaired by mutagenic end-joining or HR would generate an ~5000 bp product, while amplification of a locus repaired by SSA would generate a distinct ~1300 bp fragment. Indeed, upon targeting Cas9 to DSB-Spectrum_V2, a 1300 bp PCR product was detected (Fig. 3d), and the sequence of this PCR product aligned to the predicted SSA repair product (Supplementary Fig. 6a). Importantly, the abundance of this SSA repair product was markedly increased following NU7441 treatment (Fig. 3d). Hence, inhibition of DNA-PKcs reduces c-NHEJ, which is then compensated for by both an increase in HR, and SSA.

## DSB-Spectrum_V3 is a multi-pathway DSB-repair reporter that simultaneously reports mutagenic end-joining, SSA, and HR

Based on the above results, we modified DSB-Spectrum_V2 to generate a new reporter capable of distinguishing between DSB-repair by HR, mutagenic end-joining, or SSA (Fig. 4a). This was achieved by inserting a pgk-promoter controlled mCherry gene in place of the pgk-Puro gene in the region separating BFP and GFP in DSB-Spectrum_V2 (Fig. 4b). With this reporter, named DSB-Spectrum_V3, repair by mutagenic end-joining will result in loss of BFP, but not mCherry

expression, and can therefore be distinguished from SSA which will result in loss of both fluorescent proteins (Fig. 4a, b). A single copy of DSB-Spectrum_V3 was integrated into the genome of HEK 293T cells (Supplementary Fig. 2a, b), and a single clone was expanded, which was GFP⁻, BFP⁺, and mCherry⁺, consistent with the design of the reporter. Similar to what was observed for DSB-Spectrum_V2 cells, targeting of Cas9 to DSB-Spectrum_V3 resulted in the appearance of a GFP⁺ and BFP⁻ population (Fig. 4c, d, compare AAVS1sg versus BFPsg). The BFP⁻ population could be further divided into an mCherry⁺ and mCherry⁻ population, reflecting DSB-repair by mutagenic end-joining and SSA, respectively (Fig. 4c, d). As previously done for DSB-Spectrum_V2 (Supplementary Fig. 4a), we monitored the kinetics of BFP and mCherry loss to determine the incubation time after Cas9 transfection that would generate optimal separation between the fluorescent populations. This revealed that the BFP⁻, mCherry⁺ population could be distinguished from the BFP⁻, mCherry⁻ population at 72 h after Cas9 transfection, although even better separation was achieved when extending the incubation time to 96 h (Supplementary Fig. 6b). In this DSB-Spectrum_V3-expressing cell-line, a surprisingly large fraction of the cells lost both BFP and mCherry expression (Fig. 4c, d). To validate that the BFP⁻, mCherry⁻ population was indeed the consequence of repair by SSA, we depleted the known SSA-factor Rad52 by RNAi. This resulted in a significant reduction in the frequency of BFP⁻, mCherry⁻ cells (Fig. 4e, f), demonstrating that DSB-Spectrum_V3 reports on SSA through loss of mCherry fluorescence.

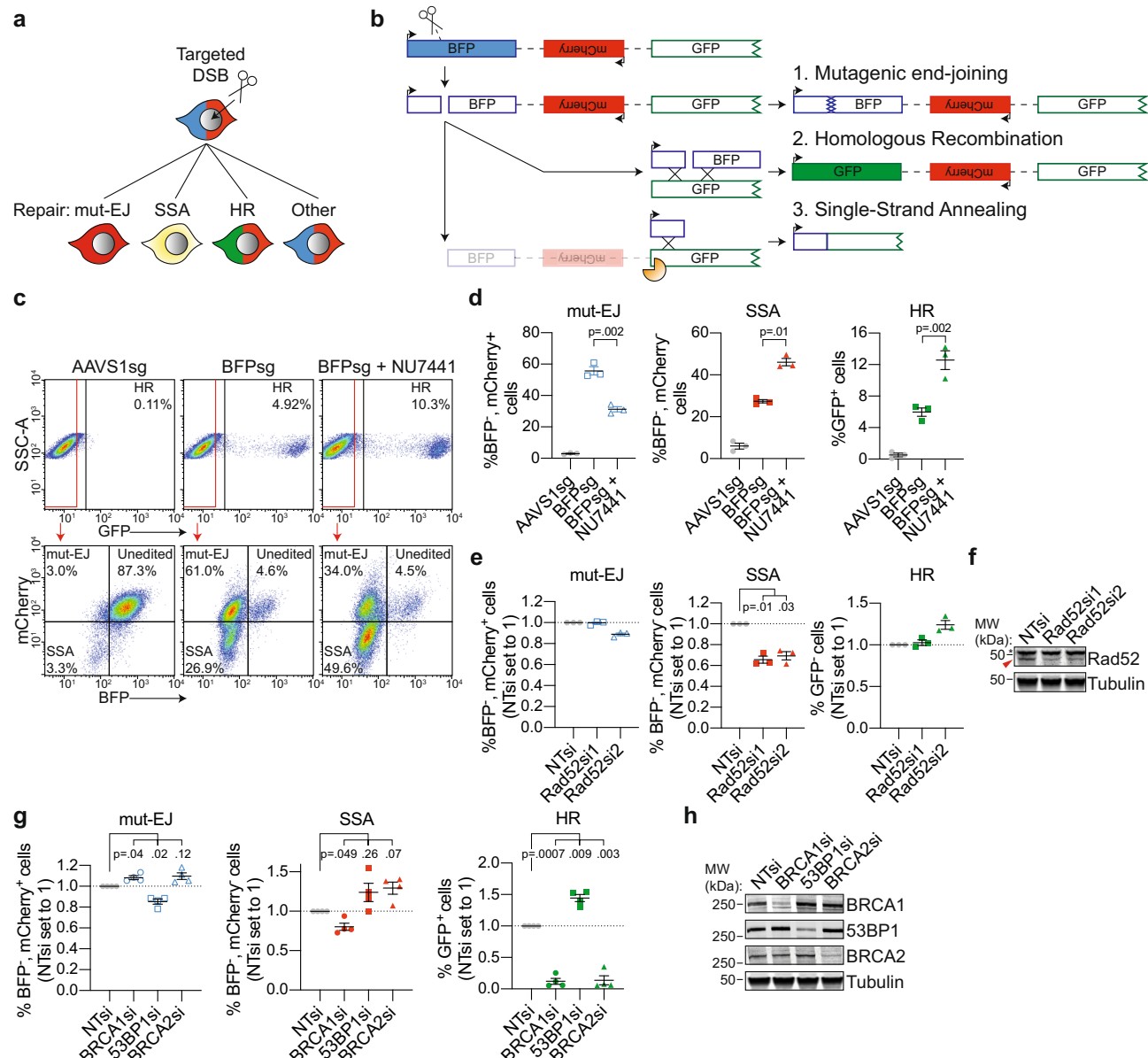

**Fig. 4 | DSB-Spectrum_V3 is a reporter for mutagenic end-joining, SSA, and HR.**
**a** Cartoon depicting potential outcomes of a multi-pathway reporter cell-line designed to quantify DSB-repair by mutagenic end-joining (mut-EJ), SSA, and HR. **b** Diagramatic representation of the genomic DSB-repair reporter construct DSB-Spectrum_V3. Scissors indicate the Cas9-target site, orange pacman indicates endogenous nucleases. **c, d** DSB-Spectrum_V3 cells were transfected with Cas9 and an sgRNA targeting either AAVS1 or BFP, followed by treatment with NU7441 (2 μM). At 72 h after Cas9 transfection cells were analyzed by flow cytometry. Panel **c** shows representative flow plots. Panel **d** shows quantification of the three repair pathways from multiple experiments

($n = 3$; mean ± SEM; ratio paired $t$-test, two-tailed). **e** DSB-Spectrum_V3 cells were transfected with indicated siRNAs, followed by flow cytometric analysis of mut-EJ, SSA, and HR ($n = 3$; mean ± SEM; One-way ANOVA, post-hoc Dunnett's). **f** Western blot of lysates from cells analyzed in panel **e**. Tubulin is used as a loading control. Red arrow indicates Rad52 band, asterisk indicates nonspecific background band. **g** Mut-EJ, SSA, and HR was analyzed as in panel **e**, following siRNA-mediated knockdown of the indicated repair factors ($n = 4$; mean ± SEM; One-way ANOVA, post-hoc Dunnett's). **h** Western blot of lysates from cells analyzed in panel **g**. Source data for panels **d**, **e**, and **g** are provided as a Source Data file.

Next, we used this reporter to further examine the effects of DNA-PKcs inhibition on DSB-repair pathway choice. Treatment with NU7441 did not affect the frequency of BFP loss, but markedly increased the frequency of mCherry⁻ cells, at the expense of mCherry⁺ cells, within the BFP⁻ population (Fig. 4c, lower panel). Quantification of multiple experiments with DSB-Spectrum_V3 cells demonstrated that NU7441 treatment increased repair by both HR and SSA while markedly inhibiting repair by mutagenic end-joining (Fig. 4d). To test whether this phenotype was caused by on-target inhibition of DNA-PKcs, rather than by potential off-target effects of NU7441, we repeated the DSB-Spectrum_V3 experiment with AZD7648 and M3814, two other, more recently identified inhibitors of DNA-PKcs[35,36]. These treatments

resulted in a similar shift in DSB-repair pathway choice from end-joining to SSA and HR (Supplementary Fig. 7a), demonstrating that this phenotype was indeed caused by on-target inhibition of DNA-PKcs.

The unique capacity of DSB-Spectrum V3 to report on three different DSB-repair pathways simultaneously provides a method to assess competition between end-joining, HR, and SSA repair for the same DSB substrate. We therefore made use of the DSB-Spectrum_V3 cell-line to examine crosstalk between repair pathways following perturbation of known HR and end-joining factors. First, we depleted BRCA1, 53BP1 or BRCA2 by RNAi and quantified mutagenic-EJ, SSA, and HR by flow cytometry. As expected, based on its function as an end-resection inhibitor[5], 53BP1 depletion reduced the percentage of BFP⁻,

mCherry⁺ cells, but increased the percentage of both BFP⁻, mCherry⁻ cells, and GFP⁺ cells, indicating suppression of mut-EJ and enhanced SSA and HR (Fig. 4g, h). Furthermore, the DSB-Spectrum_V3 reporter revealed that depletion of BRCA1 and BRCA2 strongly reduced HR and resulted in a mild promotion of mut-EJ, but had a differential effect on SSA (Fig. 4g, h). Whereas BRCA1-loss reduced SSA, depletion of BRCA2 promoted SSA (Fig. 4g, h). This finding is consistent with what has been published using a combination of individual single-pathway reporters[24], and fits well with a model in which the end-resection function of BRCA1 promotes both HR and SSA, while the HR-specific Rad51-loading function of BRCA2 following end-resection directs repair towards HR, and away from SSA[37]. Notably, these data can also explain some seemingly puzzling results that were obtained with HEK 293T DSB-Spectrum_V2 cells. While knockdown of the end-joining factor 53BP1 resulted in suppression of mutagenic repair (fewer BFP⁻ cells) and enhancement of HR (more GFP⁺ cells; Supplementary Fig. 7b, c), as expected, knockdown of BRCA1 by siRNA markedly suppressed HR, but surprisingly reduced, rather than enhanced, mutagenic repair, as evidenced by the reduction in both GFP⁺ and BFP⁻ cells, respectively (Supplementary Fig. 7b, c). The reduced BFP loss that we observed after knockdown of BRCA1 in the DSB-Spectrum_V2 cells can be reconciled as potentially arising from a reduction in mutagenic repair by SSA when BRCA1 is not present.

Taken together, these data validate the utility of DSB-Spectrum_V3 as a multi-pathway reporter that simultaneously quantifies repair by mutagenic end-joining, SSA, and HR, with the ability to reveal both expected and unexpected DSB-repair pathway interactions.

### DSB-repair by SSA is frequent in multiple reporter contexts, genomic contexts, and cell-lines

The high frequency of DSB-repair by SSA in DSB-Spectrum_V3 cells (~27%, Fig. 4d) suggests that it can be a dominant pathway of repair, even in the presence of functional c-NHEJ and HR. To validate that this high frequency of SSA was not an artifact of the specific clonal DSB-Spectrum_V3 cell-line used, we repeated the experiments with a second monoclonal HEK 293T cell-line. A similar frequency of DSB-repair by SSA was observed in this second DSB-Spectrum_V3 clone, which further increased upon treatment with NU7441, demonstrating that the observed phenotypes are not clone-specific (Supplementary Fig. 7d).

To validate the generality of the high frequency of SSA, we further quantified DSB-repair by SSA in DSB-Spectrum_V2 cells. In this reporter, the homologous regions are separated by a different sequence than in DSB-Spectrum_V3, but more importantly, for this clonal cell-line we had confirmed that the reporter construct was integrated into a different genomic location than in the DSB-Spectrum_V3 clone used in Fig. 4 (Supplementary Fig. 2a, b). In DSB-Spectrum_V2, SSA repair of the DSB in the BFP gene will result in loss of the pgk-Puro cassette (Fig. 3c). SSA can therefore be quantified by measuring the number of cells that have lost puromycin resistance after generating a DSB in the reporter. We transfected HEK 293T DSB-Spectrum_V2 cells with either a control (AAVS1sg) of BFP-targeting sgRNA, and allowed gene editing to occur for 72 h. Next, transfected cells were sorted by FACS, and either analyzed by flow cytometry to determine the frequency of mutagenic repair and HR, or plated to determine clonogenic survival in the absence or presence of puromycin. Using this experimental setup, we observed that ~70% of the DSB-Spectrum_V2 cells lost BFP expression and ~4.5% underwent repair by HR (Supplementary Fig. 7e). Moreover, loss of puromycin resistance was specifically observed in the BFPsg-transfected cells. On average, 22.3% of the cells lost puromycin resistance upon targeting Cas9 to DSB-Spectrum_V2, suggesting that this fraction of cells lost the pgk-Puro cassette due to DSB-repair by SSA (Supplementary Fig. 7e). Notably, this frequency of SSA is very similar to the frequency of SSA detected by mCherry loss in DSB-Spectrum_V3 cells (Fig. 4d).

Finally, to test whether SSA is also a frequently employed pathway in other cell-lines than HEK 293T, we performed the same assay in DSB-Spectrum_V2 U2OS cells. This revealed that around 20% of cells lose puromycin resistance upon targeting Cas9 to the reporter (Supplementary Fig. 7f). When compared to the 40% of total cells that had lost BFP expression and the ~0.4% of cells that underwent repair by HR (Supplementary Fig. 7f), these data indicate that, also in U2OS cells, a substantial fraction of DSBs are repaired by SSA. Taken together, these results demonstrate that SSA-mediated repair of a DSB in DSB-Spectrum can be frequent in a variety of genomic contexts and in multiple cell-lines.

### DSBs at endogenous genomic loci can be repaired by SSA through annealing of homologous Alu elements or homologous gene regions

The DSB-Spectrum reporters are potentially prone to SSA-mediated repair due to the highly homologous BFP and GFP sequences within 3 kb of the DSB (Fig. 3c). We therefore investigated whether SSA is also employed to repair DSBs in endogenous genomic loci. It has been suggested that Alu elements can anneal during DSB-repair by SSA[9], because these 200–300 bp elements are abundant in the human genome and can share high levels of sequence homology[38]. To test this hypothesis, we designed sgRNAs to target Cas9 to four unique loci in the genome, located in the *FANCA*, *BRCA1*, *BTK*, or *SPAST* gene (Fig. 5a, b). At all four loci Cas9 was targeted to an Alu element (Alu_1) that shares considerable homology (≥84%) with a second Alu element (Alu_2) located at a distance between ~4.8 and 7.6 kb away (Fig. 5a, b). We hypothesized that upon Cas9-induced DSB generation, long-range end-resection could expose both homologous Alu elements, which in turn could anneal during SSA, followed by removal of the region between the Alu elements. The presence of such an SSA-induced deletion product could then be detected by PCR (Fig. 5a). We transfected HEK 293T DSB-Spectrum_V2 cells with Cas9 cDNA and an sgRNA, to target a negative control locus (AAVS1), a positive control locus (the BFP gene in DSB-Spectrum), or the Alu_1 element in the *FANCA, BRCA1, BTK, or SPAST* gene. We next isolated genomic DNA, and performed PCR to detect the deletion product generated by annealing between the Alu elements during SSA-mediated repair of the target loci (Fig. 5b). For all loci, we could readily detect a specific PCR product of the expected size, only when cells were transfected with the sgRNAs targeting the Alu_1 element in *FANCA, BRCA1, BTK* or *SPAST*, but not when they were transfected with the AAVS1-targeting control sgRNA (Fig. 5c, d). Sequencing of these PCR products confirmed that they corresponded to the predicted products of SSA repair between the Alu elements (Supplementary Fig. 8a, d). Moreover, upon knockdown of the SSA-factor Rad52, the yield of these PCR products was reduced, confirming that they were the consequence of SSA-mediated repair of the Cas9-induced DSB (Fig. 5c–e). These results show that Alu elements, of which there are over a million interspersed throughout the human genome[38], can drive DSB-repair by the efficient but highly mutagenic SSA pathway.

Our prior results with DSB-Spectrum reporters indicated that inhibition of DNA-PKcs strongly promotes SSA (Figs. 3d and 4d). To test whether this is also the case for SSA-mediated repair of endogenous genomic loci, we targeted the four Alu-containing loci as described above but now included treatment with NU7441 for the duration of gene editing. The SSA repair products detected upon DSB generation in the four Alu-containing target loci, as well as in the BFP-positive control locus, were between 1.6-fold (*SPAST*) and 4.6-fold (*BTK*) more abundant when DNA-PKcs was inhibited by NU7441 treatment (Fig. 5f, g). Interestingly, for the *FANCA* and *BTK* loci, a second, smaller PCR product was observed in addition to the predicted SSA repair product, and this smaller-sized product was also increased in abundance upon NU7441 treatment (Fig. 5f, g). Both of these loci also contain a second homologous Alu element downstream of the Alu_2 element, to which the resected Alu_1 could also anneal

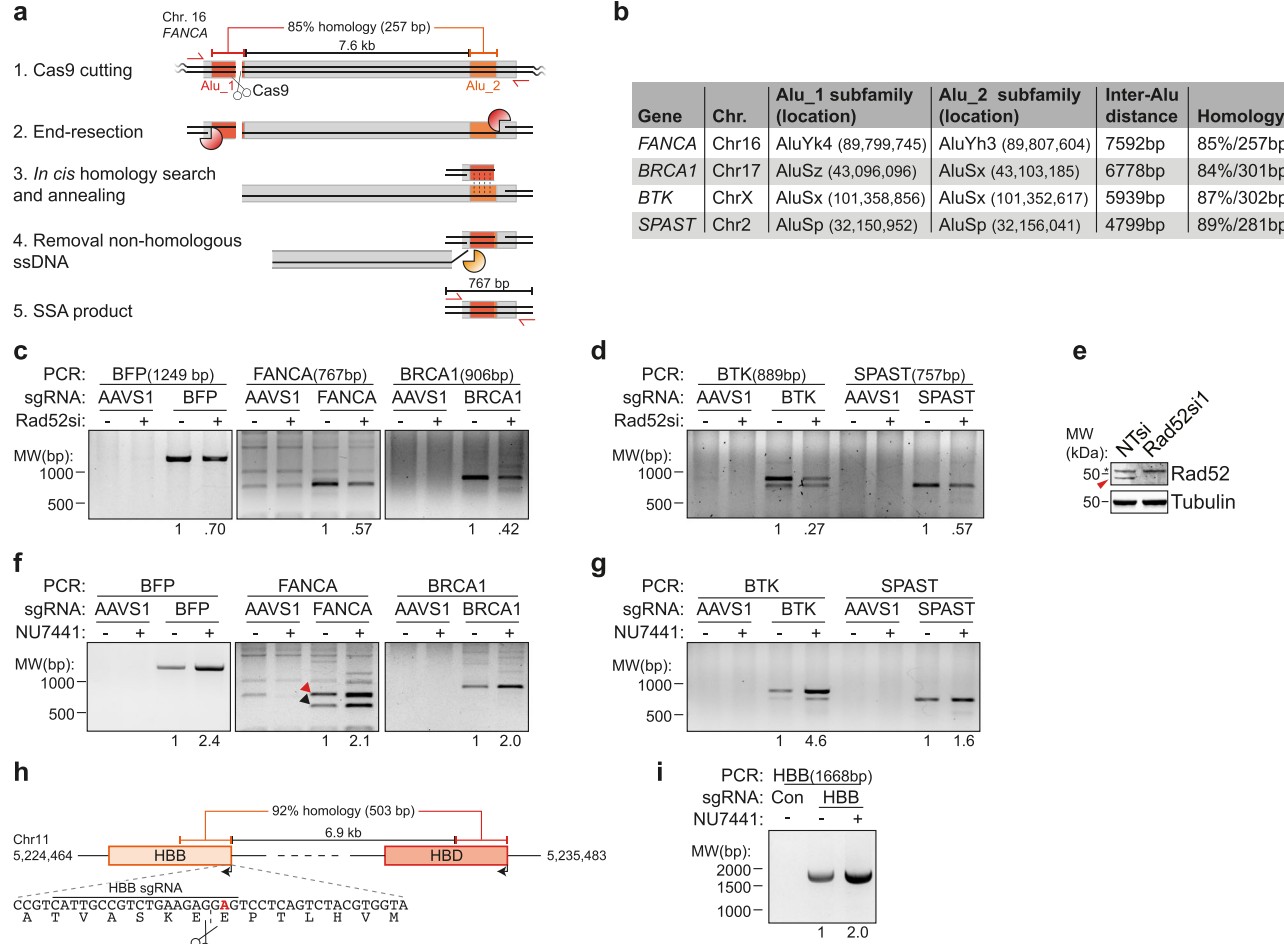

**Fig. 5 | SSA contributes to repair of DSBs in endogenous genomic loci.**
**a** Schematic representation of SSA repair of a Cas9-induced DSB at the Alu_1 target site in the *FANCA* gene. Indicated are the primers (in red) to amplify the SSA repair product, and the nucleases that perform end-resection (orange pacman) or ssDNA flap removal (yellow pacman). **b** Table indicating the subfamily, chromosomal location, and level of sequence homology of the Alu elements that could anneal during SSA repair of the DSB at the Alu_1 target loci. **c**, **d** HEK 293T DSB-Spectrum_V2 cells (**c**) or regular HEK 293T cells (**d**) were transfected with a nontargeting control siRNA ("–" lanes) or Rad52-targeting siRNA, then retransfected with the indicated Cas9-sgRNA constructs. Genomic DNA was isolated and SSA DSB-repair products PCR amplified. Representative gel images are shown (*n* = 2). Expected size of the SSA repair product and the quantified SSA band intensities are shown at the top and the bottom of the image, respectively. **e** Western blot of lysates from panels **c** and **d**. Red arrow indicates Rad52, asterisk indicates nonspecific band. **f**, **g** As in panels **c** and **d**, but excluding siRNA transfection, and including NU7441 treatment (2 µM). In the FANCA panel, arrows indicate the SSA products between the Alu_1 element and either the Alu_2 element (red), or a more downstream AluSx element (black). Representative gel images are shown (*n* = 2). **h** Schematic showing the genomic location and level of homology between the *HBB* and *HBD* genes. Zoom shows DNA and amino acid sequence at the start of the *HBB* gene (coding strand). Indicated are the sequence recognized by the HBB-targeting sgRNA, and the sickle cell disease-causing A > T mutation (bold red). **i** HEK 293T cells were transfected with Cas9 and either a control sgRNA or an HBB-targeting sgRNA. Cells were treated with NU7441 (2 µM) or vehicle control and the SSA DSB-repair products were PCR amplified from genomic DNA. The expected size of the SSA repair product is shown at the top, quantification of SSA band intensities is shown at the bottom.

during SSA, resulting in a larger deletion and thus a smaller PCR product. We tested this for the *FANCA* locus by sequence analysis of the smaller PCR product, which indeed demonstrated it to be the consequence of SSA between the Alu_1 element and an AluSx element directly downstream of the Alu_2 element (Supplementary Fig. 8a). Together, these data demonstrate that DSBs in endogenous genomic loci can be repaired by SSA between homologous Alu elements and that this mutagenic repair is strongly stimulated by inhibition of DNA-PKcs kinase activity.

To explore DSB-repair by SSA in the human genome outside the context of Alu elements, we focused on the *HBB* gene that encodes the hemoglobin subunit β-globin. A point mutation in this gene that results in a Glu7Val amino acid substitution is the single cause of sickle cell disease (SCD), a severe hematological disorder[39]. Intense research efforts are ongoing to develop gene editing therapeutics, Cas9-based and others, that can cure the patient by correcting the

SCD point mutation in patient-derived hematopoietic stem cells[40]. Notably, a 503 bp region in the *HBB* gene that includes the site mutated in SCD is 92% homologous to a region in the *HBD* gene, which is located ~7 kb downstream on the same chromosome (Fig. 5h). Repair of a DSB in this *HBB* gene region could therefore occur by SSA between the *HBB* and *HBD* genes. To test this hypothesis, we targeted Cas9 to the *HBB* locus, generating a DSB close to the site mutated in SCD, and assessed the presence of an HBB/HBD SSA repair product by PCR. A specific SSA repair product could be detected in cells transfected with the HHB sgRNA, but not in cells transfected with a control sgRNA (Fig. 5i). This demonstrates that SSA contributes to the repair of nuclease-induced DSBs in the *HBB* locus through recombination with the *HBD* gene. Furthermore, inhibition of DNA-PKcs strongly promoted SSA repair of the *HBB* gene (Fig. 5i). Thus, SSA in the human genome is not limited to Alu elements and can also occur between other homologous gene regions.

## Inhibition of long-range end-resection promotes HR and reduces SSA

Our data indicate that SSA contributes to the repair of Cas9-induced DSBs in endogenous genomic loci, including Alu elements and the *HBB* gene, a locus that is under investigation as a target in gene editing therapy development. Genome editing approaches that depend either on end-joining to generate knock-outs or on HR to introduce designed mutations would therefore benefit from methods that deviate repair away from SSA towards these other pathways. This is particularly challenging for HR because it shares the initial repair steps with SSA. It is unclear how the branching between these pathways is regulated. Therefore, we aimed to identify factors that regulate DSB-repair pathway choice between SSA and HR. Data from Ochs et al. indicated that hyper-resection may promote SSA over HR[41]. We therefore hypothesized that there is a differential requirement for specific end-resection factors. To study this, we transfected DSB-Spectrum_V3 cells with siRNAs to deplete either the short-range end-resection factors Mre11 or CtIP, or the long-range end-resection factors Exo1 or DNA2, and analyzed the modes of DSB-repair by flow cytometry. As shown in Fig. 6a, b, loss of Mre11 and CtIP resulted in a reduction of DSB-repair by both SSA as well as HR, consistent with a requirement for end-resection for both pathways (Fig. 6a, b). A similar reduction in SSA was observed upon knockdown of Exo1 and DNA2, but surprisingly, loss of these factors promoted, rather than reduced, HR (Fig. 6a–c). Thus, the efficiency of DSB-repair by SSA, but not HR, is dependent on factors involved in long-range end-resection, and these factors can even inhibit HR.

This finding prompted us to test, using DSB-Spectrum_V3 cells, whether knockdown of either Exo1 or DNA2 would increase HR, while also limiting the high levels of SSA induced by DNA-PKcs inhibition. We found that DNA-PKcs inhibition alone promoted HR (compare gray and colored symbols), and this increase was further potentiated by depletion of Exo1 or DNA2 (Fig. 6d right panel). Furthermore, Exo1 depletion reduced the frequency of repair by SSA, both in untreated cells as well as in NU7441-treated cells (Fig. 6d middle panel). This reduction was sufficient to limit the percentage of SSA in cells co-treated with both NU7441 and Exo1 siRNA, to a level comparable to the percentage of SSA observed in untreated cells (Fig. 6d, middle panel, colored symbols). These findings indicate that in DSB-Spectrum_V3, DSB-repair pathway choice can be efficiently channeled towards HR without promoting mutagenic SSA, by concomitantly inhibiting both c-NHEJ and long-range end-resection by Exo1. Notably, in untreated cells, depletion of DNA2 resulted in a less substantial reduction of repair by SSA compared to Exo1 depletion (Fig. 6a, d), and consequently, depletion of DNA2 did not limit the high frequency of SSA following NU7441 treatment (Fig. 6d).

To validate this function of Exo1 in SSA at endogenous genomic sites, we assessed repair by SSA of DSBs in the *BTK*, *SPAST*, and *HBB* loci. Exo1 expression was silenced by RNAi, followed by transfection of Cas9-sgRNA constructs and subsequent detection of SSA by PCR analysis, as described above. RNAi-mediated depletion of Exo1 substantially reduced the frequency of DSB-repair by SSA in all of these endogenous loci, similar to what was found for DSB-Spectrum_V3 (Fig. 6e–h). Next, to study HR, we designed an exogenous dsDNA repair template that introduces a T7 primer binding site at the DSB-repair junction in the *BTK* gene (Fig. 6i). This template was co-transfected with the Cas9-sgRNA construct, and subsequently, HR could be monitored by a PCR strategy designed to specifically amplify the repaired locus containing the T7 primer binding site (Fig. 6i). Validating the experimental design, a specific HR repair product was detected when Cas9, the BTK-targeting sgRNA, and a repair template were introduced into the cells, whereas no HR product was detected upon transfection of a control sgRNA or in absence of the template (Fig. 6j). Interestingly, depletion of Exo1 did not affect the yield of this HR product, suggesting that Exo1 neither inhibits, nor promotes, HR in this experimental setting using an exogenous template (Fig. 6j).

The differential effect of Exo1 loss on HR in the *BTK* locus compared to the DSB-Spectrum locus could be explained by a locus-specific role of Exo1. Alternatively, the difference in HR repair templates could provide an explanation, as HR at the *BTK* locus was examined using an ectopic repair template, whereas repeat-containing reporters like DSB-Spectrum predominantly recombine with the sister chromatid in the context of chromatinized DNA[42]. Because the latter is more reflective of DSB-repair in normal physiology, we aimed to study sister chromatid templated HR in a context other than DSB-Spectrum. We focused on the *HBB* locus because we reasoned that repair of the DSB in the *HBB* gene could also occur by HR, in addition to SSA (described above), using the downstream *HBD* gene as a template, similar to what occurs between the BFP and GFP genes in DSB-Spectrum (Fig. 6k). This would result in several mutations at the *HBB* locus, that could be detected by sequence analysis of the DSB-repair junction, and quantified by TIDER[31]. To test this, we targeted Cas9 to the *HBB* locus and could indeed detect HBB to HBD gene conversion at the DSB junction, albeit at very low frequencies (Fig. 6l). Importantly, depletion of Exo1 increased the frequency of these HR repair products (Fig. 6l). HR at the *HBB* locus was also strongly enhanced by NU7441 treatment, and this was even further promoted when combined with Exo1 knockdown (Fig. 6l). Thus, similar to what we found for the DSB-Spectrum locus, HR at the *HBB* locus is inhibited by Exo1 expression. Taken together, the data indicate that at a Cas9-induced DSB, long-range end-resection can determine pathway choice between SSA and HR with the sister chromatid.

## Discussion

In this manuscript, we describe and validate three distinct reporter systems for DNA DSB-repair, DSB-Spectrum_V1, V2, and V3, capable of flow cytometric quantification of repair by error-free c-NHEJ versus HR in the case of V1, mutagenic repair versus HR in case of V2, or mutagenic end-joining, versus HR versus SSA in case of V3. These DSB-Spectrum reporters build on a large history of DSB-repair reporter constructs, and their design is based on commonly used single-pathway reporter systems such as DR-GFP and variations there-of[23,25]. These single-pathway reporters, designed by others, have proven extremely valuable in DSB-repair research[27]. The DSB-Spectrum reporters that we describe here have the added advantage that they can simultaneously quantify the frequency of DSB-repair by multiple different pathways within a single population. This facilitates the analysis of pathway crosstalk and reveals DSB-repair pathway interdependence and compensation as they compete for the same DSB substrate.

A number of fluorescent two-pathway reporters have been described[27], most of which require the addition of an exogenous repair template to study HR. In contrast, DSB-Spectrum does not require an exogenous template to quantify HR while simultaneously reporting on either error-free or error-prone end-joining. Furthermore, DSB-Spectrum_V3 can report on three, rather than two, DSB-repair pathways, which is only shared with the SSA-TLR reporter developed by Certo, Scharenberg, and colleagues[43]. That reporter can quantify mutagenic end-joining, SSA and HR, like DSB-Spectrum_V3. However, the frequency of detected end-joining events by the SSA-TLR reporter is very low compared to DSB-Spectrum, because the SSA-TLR can only detect these events if the resulting InDels shift the reading frame by two bp[43]. Finally, in the case of the SSA-TLR, the detected frequency of HR events is strongly dependent on the amount of repair template co-delivered with the nuclease, and in the majority of assays presented by the authors, the HR frequency is lower than 0.5%[43]. Thus, we believe that the technical simplicity, high efficiency, and repair pathway inclusivity make DSB-Spectrum reporters ideal tools to study DSB-repair on the network level.

Our finding that in both HEK 293T and in U2OS cells, the frequency of detected DSB-repair by SSA was substantially higher (20–25%) than the frequency of detected repair by HR (0.5–5%) is

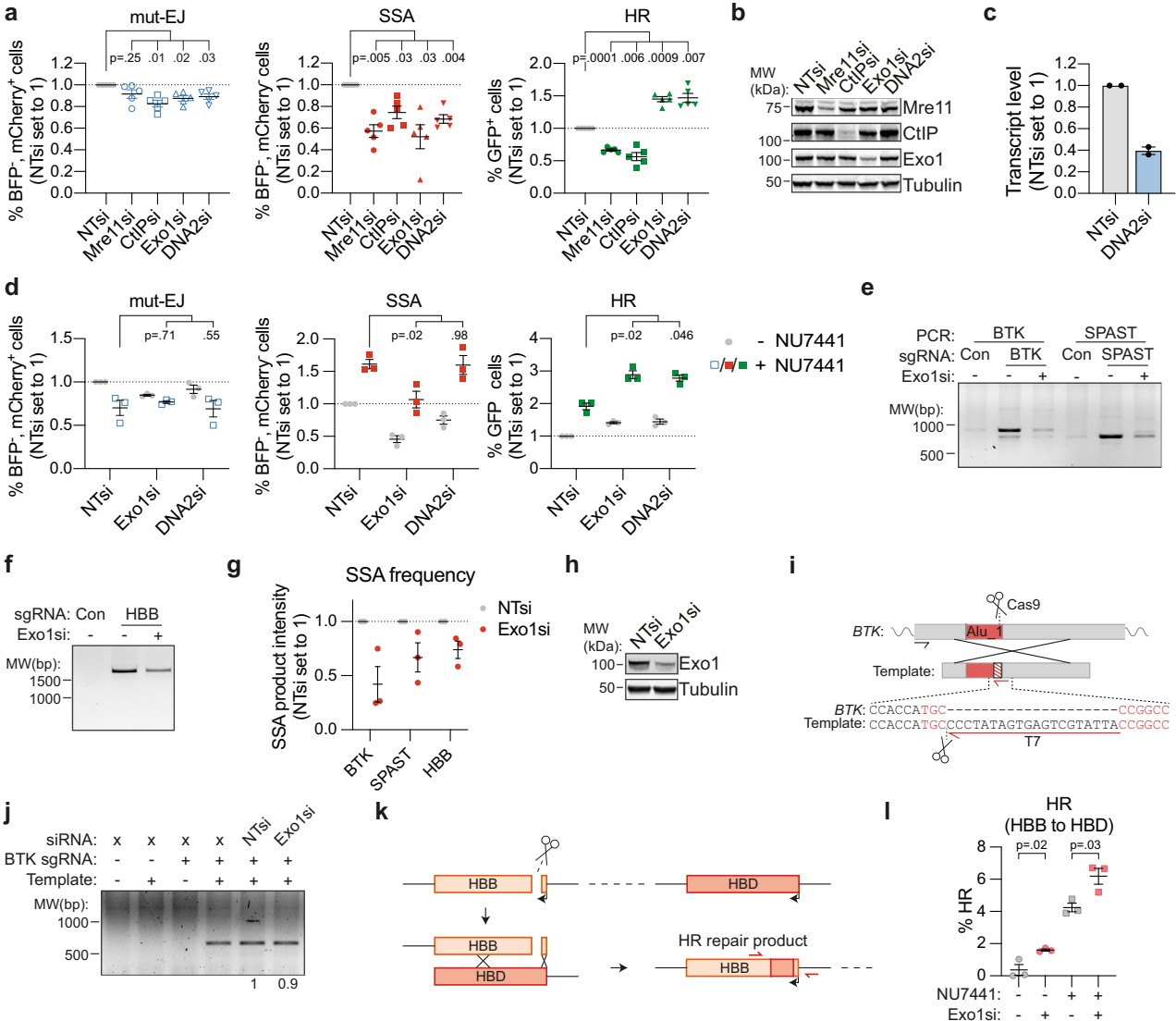

**Fig. 6 | The long-range end-resection factors Exo1 and DNA2 are required for SSA, but inhibit HR. a** DSB-Spectrum_V3 cells were transfected with indicated siRNAs, followed by transfection with Cas9 and an sgRNA targeting a control locus or BFP. Next, cells were analyzed by flow cytometry, and repair pathways quantified as in Fig. 1e (*n* = 5; mean ± SEM; One-way ANOVA, post-hoc Dunnett's). **b** Western blot of lysates from cells analyzed in panel **a**. **c** Quantification by qRT-PCR of DNA2 knockdown in cells used in panel **a**. Plotted are the mean and SEM of two independent qPCR experiments each done with a different primer pair. **d** As in panel **a**, in cells treated with or without NU7441 (2 μM; *n* = 3; mean ± SEM; One-way ANOVA, post-hoc Dunnett's). **e**–**g** HEK 293T cells were transfected with a non-targeting control siRNA ("-" lanes) or Exo1 siRNA, then retransfected with indicated Cas9-sgRNA constructs (Con = AAVS1-targeting). SSA DSB-repair products were PCR amplified from genomic DNA. Panels **e** and **f** show a representative gel image, panel **g** shows the quantification (*n* = 3; mean ± SEM). **h** Western blot of lysates from cells analyzed in panel **e**. **i** Cartoon depicting the HR strategy at the *BTK* locus. The black and red primers were used to amplify HR repair products. Sequence in red font indicates part of the sgRNA target region. **j** As in panel **e**, but including transfection of a repair template. The HR DSB-repair product was PCR amplified, rather than the SSA DSB-repair product. A representative gel image from three independent experiments is shown; numbers at the bottom show quantification of HR band intensities. **k** Cartoon depicting HR repair of the *HBB* locus using the downstream *HBD* gene as a repair template. Primers used to amplify the repair junction for TIDER analysis are depicted in red. **l** HEK 293T cells were transfected with an HBB sgRNA, and the frequency of HBB to HBD gene conversion products quantified by TIDER analysis (*n* = 3; mean ± SEM; unpaired *t*-test, two-tailed). Source data for panels **a, c, d, g,** and **l** are provided as a Source Data file.

surprising but entirely consistent with the high frequencies of SSA that have been reported using another reporter systems[11,43–45]. Of note, the majority of DSB-repair reporters, including DSB-Spectrum, contain highly homologous sequences adjacent to the DSB. This generates an optimal sequence context for SSA and may therefore predispose reporters to repair by this pathway. This SSA-prone sequence context, however, might not be rare in the human genome, which contains a large number of repetitive sequences[46], including over a million Alu elements[38]. Consistent with this notion, we show that SSA between adjacent Alu elements contributes to DSB-repair and causes large deletions in the *BRCA1, BTK, FANCA,* and *SPAST* genes. Each of these four genes is mutated in a hereditary disease/syndrome, which are the hereditary breast and ovarian cancer syndrome (*BRCA1*), α-gammaglobulinemia (*BTK*), Fanconi anemia (*FANCA*), and spastic paraplegia 4 (*SPAST*). Interestingly, for each of the four genes, patients or mutant carriers have been described that carry deletions between the exact Alu elements that we found had recombined during SSA repair[47–51]. The molecular mechanism responsible for the Alu–Alu deletions seen in the familial *BTK, BRCA1, FANCA,* or *SPAST* mutant genes is not known, but our findings indicate that SSA-mediated repair of a DSB in the vicinity would have the potential to generate such deletion products.

Other homologous regions besides Alu elements could also drive SSA, as we show here for the *HBB* and *HBD* genes. Hence, our findings indicate that SSA can also contribute to the repair of loci that are of interest for clinical genome editing. Importantly, given the abundance of Alu elements and homologous gene families, repair by SSA should be monitored when assessing gene editing outcomes. Notably, SSA repair products are generally missed by next-generation sequencing of a small region surrounding the DSB junction[52]. This is exemplified by recent studies that used long-read sequencing and PCR techniques and found previously undetected kilobase-sized genomic deletions at Cas9-target sites, including those in genetically modified human embryos[52–54]. It would be interesting to examine whether SSA is responsible for these large deletions.

Our studies also demonstrated that DSB-repair by SSA is strongly promoted by inhibition of DNA-PKcs kinase using three different inhibitors, consistent with a slight increase in SSA that was reported upon treatment with NU7026, another DNA-PKcs kinase inhibitor[55]. This SSA-promoting effect is important because inhibition of DNA-PKcs has been used to increase the HR to c-NHEJ ratio in the context of CRISPR-mediated genome editing[8,56,57]. To provide a potential solution for the high levels of SSA following inhibition of end-joining, we used DSB-Spectrum_V3 cells to study the DSB-repair pathway crosstalk between end-joining, HR, and SSA. We found that optimal SSA requires the long-range end-resection factors Exo1 and DNA2, as anticipated on basis of the 3 kb distance between the homologous regions in DSB-Spectrum_V3. This is substantially larger than the short-range resection span of 200–300 nucleotides typically generated by Mre11 and CtIP[58]. To our surprise, our studies showed that loss of Exo1 did not reduce HR when using an ectopic repair template at the *BTK* locus, and even promoted HR at the DSB-Spectrum_V3 and *HBB* loci. This contradicts the conventional view of end-resection and pathway choice, suggesting that Exo1 and DNA2 are positive regulators of HR[59]. However, our data are in good agreement with published reports showing an increase in HR upon DNA2 knockdown and a decrease in HR upon hyperactivation of Exo1[60,61]. Furthermore, consistent with our findings, Chen et al. showed that reconstitution of Exo1 knock-out cells with Exo1 cDNA reduced HR, especially between repeats with some divergence in sequence[62]. This latter finding might also explain why Exo1 inhibited HR between the somewhat divergent sequences at the DSB-Spectrum and *HBB* loci, but did not affect HR between the *BTK* locus and the repair template with non-divergent homology arms. Lastly, depletion of 53BP1 induces hyper-resection, which correlates with a DSB-repair switch from Rad51-dependent HR to Rad52-dependent SSA[41]. Hence, our data, together with these previously published results, fits best with a model in which long-range end-resection is not required for HR and might, in some cases, direct DSB-repair towards SSA instead.

Taken together, our studies using the DSB-repair reporters DSB-Spectrum_V1 to V3 demonstrate that inhibition of DNA-PKcs promotes SSA as well as HR, and that loss of Exo1 or DNA2 can channel DSB-repair away from SSA towards HR. Multi-pathway analysis using DSB-Spectrum facilitates the generation of an inclusive model of DSB-repair pathway choice (Fig. 7) and provides a comprehensive understanding of DSB-repair outcomes following perturbation of the DSB signaling and repair network.

## Methods

### Cloning

All cloning was done using standard PCR protocols, restriction/ligation protocols, or fragment assembly using the NEBuilder HiFi DNA Assembly Master Mix (New England Biolabs) according to the manufacturer's instructions. Cloning design and sequence analysis were done using Snapgene software (from Insightful Science). DNA plasmids were purified using Plasmid Mini/Midi or Maxi kits (Qiagen) according to the manufacturer's instructions. The DSB-Spectrum_V2

construct was generated in multiple steps. First, the gene encoding for eBFP1.2 was generated by multiple rounds of site-directed mutagenesis of pEGFP-C1 (Clontech). Next, an *Xho*I restriction site was generated 5′ of the GFP gene in pDRGFP (Addgene plasmid #26475)[23], and the GFP gene was replaced by the eBFP1.2 gene using the *Xho*I/*Not*I restriction sites. Finally, the region from the start of eBFP1.2 until the end of iGFP was PCR amplified from the modified pDRGFP and ligated into *Xba*I/*Mlu*I digested pLVX-IRES-Hyg (Clontech; this procedure removes the IRES-Hygro sequence). To generate DSB-Spectrum_V1, the spacer sequence was introduced into eBFP1.2 by PCR, and wild-type eBFP1.2 in DSB-Spectrum_V2 was replaced by spacer-separated eBFP1.2 using fragment assembly. To generate DSB-Spectrum_V3, the puromycin resistance gene in DSB-Spectrum_V2 was replaced by an mCherry gene using fragment assembly. All DSB-Spectrum plasmid propagation was done in recombination-deficient bacteria (Stbl3; ThermoFisher Scientific).

To generate the Cas9/sgRNA plasmids, the puromycin resistance gene in pSpCas9(BB)−2A-Puro (PX459; Addgene plasmid #62988)[63] was replaced by either mCherry or iRFP670 using PCR and fragment assembly. A PCR strategy was designed to destroy the *Bpi*I sites in mCherry and iRFP670. Cloning of sgRNAs was done using the *Bpi*I sites in pX459 as described in Ran et al.[63]. The sgRNA protospacer sequences were as follows: AAVS1: 5′-GGCAAAATTCCCTCAGTTTA-3′, BFP sgRNA V2: 5′-ACGGGGTCCAGTGTTTTGCC-3′, BFP sgRNA 1: 5′-ACCACCCTCTCCCACGGCTA-3′, BFP sgRNA 2: 5′-GGGCAAAACACTGGACCAAT-3′, BTK sgRNA: 5′-TAAAAGGTGCAGGCCGGGCA-3′, SPAST sgRNA: 5′- TGAAAGTATATAGTAGAGCC-3′, FANCA sgRNA: 5′-GAAAGAGCCAGACTCCGTCT-3′, BRCA1 sgRNA: 5′-GGGAGGCAGACGTTGCGGAG-3′, HBB sgRNA: 5′- GTAACGGCAGACTTCTCCTC-3′. The AAVS1 sgRNA targets intron 1 of the regulatory subunit 12 C gene of protein phosphatase 1 (PPP1R12C), located on human chromosome 19. The targeting vector for DSB-Spectrum_V1, containing two sgRNAs, was generated by introducing each individual sgRNAs into a separate pX459-mCherry vector. Subsequently, the U6-sgRNA region of sgRNA 1 was put into pX459-mCherry with sgRNA 2 using XbaI/KpnI. All constructs were sequence verified by Sanger sequencing.

### Generation of DSB-Spectrum cell-lines

Cell-lines were grown at 37 °C in a humidified incubator supplied with 5% $CO_2$ in DMEM supplemented with 10% FBS. To produce lentivirus, $3 \times 10^6$ HEK 293T cells were plated in 10 cm plates and transiently transfected with pLVX-DSB-Spectrum and packaging vectors (pCMV-VSVg and pCMV-ΔR8.2) using the CalPhos mammalian transfection kit (Clontech). Virus-containing medium was filtered through a 0.45 μM filter, supplemented with PolyBrene (8 μg/ml), and transferred to a 10 cm plate with HEK 293T or U2OS target cells at 70% confluency. Due to the large size of pLVX-DSB-Spectrum the viral titer was low, so this strategy resulted in infection with low MOI (<0.2). In the case of DSB-Spectrum V1 and V2, infected cells were cultured on a selection medium containing puromycin (1 μg/ml; Invivogen). After selection, BFP⁻/GFP⁻ (V1), BFP⁺/GFP⁻ (V2), or BFP⁺/mCherry⁺/GFP⁻ (V3) cells were sorted and single-cell plated in 96-well plates by FACS, followed by expansion of single-cell clones. To test the expanded clones for efficiency of Cas9 cleavage, they were transfected with Cas9 and sgRNA(s) targeting DSB-Spectrum, and 48–96 h later, fluorescent populations were quantified by flow cytometry. Clones for which the anticipated fluorescent populations could be detected were selected and propagated as DSB-Spectrum cell-lines.

### DSB-Spectrum assays

DSB-Spectrum cells were plated and transfected the next day with pX459-Cas9-sgRNA-mCherry or pX459-Cas9-sgRNA-iRFP constructs containing either an AAVS1 sgRNA or sgRNA(s) targeting DSB-

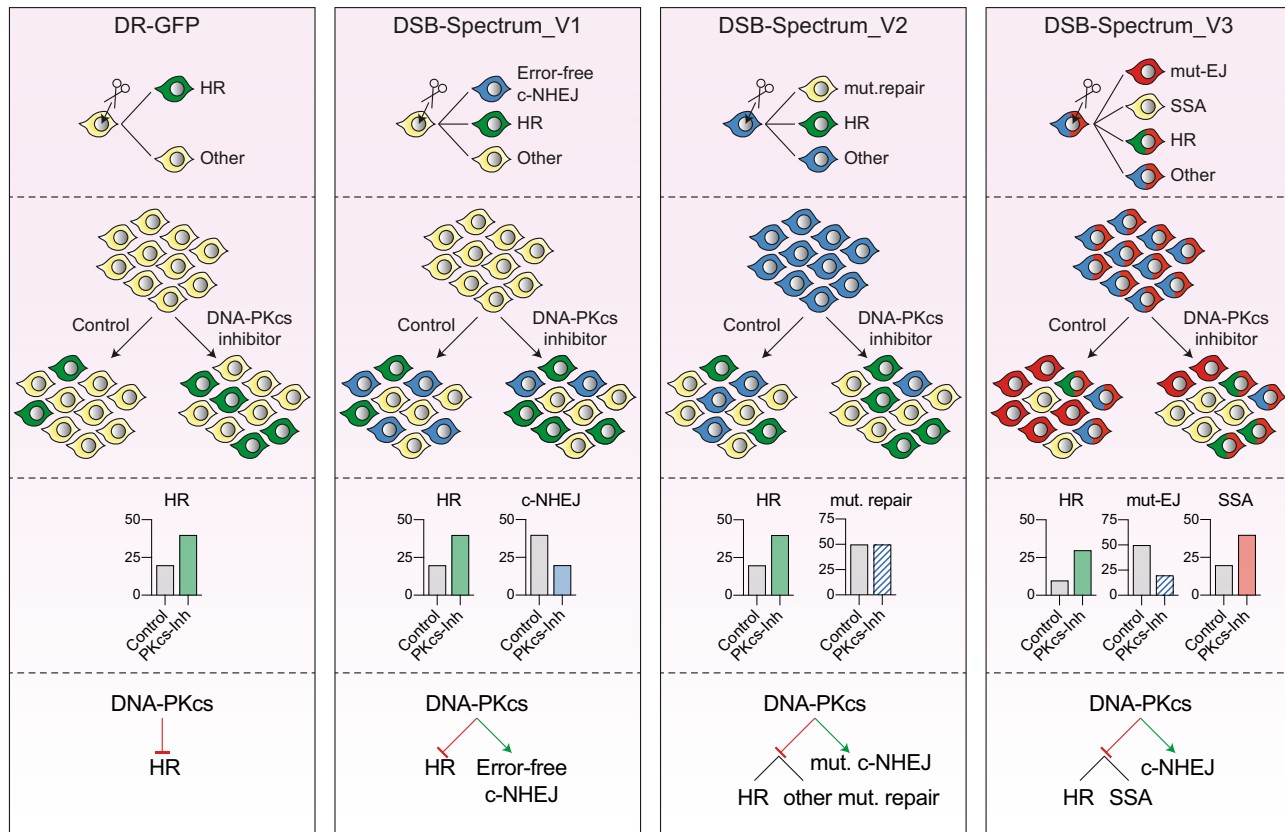

**Fig. 7 | Multi-pathway quantification of DSB-repair using DSB-Spectrum reporters allows for comprehensive analysis of repair outcomes in absence and presence of DSB-repair manipulation strategies.** Summary of the DSB-Spectrum variants described in this manuscript. The frequently used DR-GFP reporter is shown for comparison purposes. Each panel shows, from top to bottom (1) the fluorescence changes caused by repair of the site-specific DSB in the depicted reporter construct, (2) a cartoon example of a population after reporter activation, in presence or absence of DNA-PKcs inhibition as an example of a DSB-repair manipulation strategy, (3) quantification of the results in panel 2, and (4) the model resulting from the data obtained with the reporter construct.

Spectrum. Transfection was done using Lipofectamine 2000 (Invitrogen) according to the manufacturer's protocol. At 48–96 h after transfection, cells were trypsinized and analyzed by flow cytometry on a BD LSRFortessa 4 L (BD Biosciences) or a BD LSRII 4 L full (BD Biosciences) running FACS DIVA software (BD Biosciences). For experiments involving RNAi, cells were plated and transfected with siRNA the next day, followed by the replacement of the transfection medium and a second siRNA transection 24 h after the first siRNA transfection. Transfection was performed using Lipofectamine RNAiMAX (Invitrogen) according to the manufacturer's protocol. Cells were replated at 6–8 h after the second siRNA transfection and transfected with pX459-Cas9-sgRNA constructs 24 h after replating. A slightly different protocol was used for the experiments with BRCA1 and 53BP1 knockdown in HEK 293T DSB-Spectrum_V2 cells (Supplementary Fig. 3b). In this case, only one round of siRNA transfection was done, and pX459-Cas9-sgRNA constructs were transfected at 48 h after siRNA transfection. All siRNAs were Ambion Silencer Select Predesigned siRNAs (Life Technologies) with the following ID#: s11746 (Rad52si1), s11747 (Rad52si2), s224682 (BRCA1si), s14313 (53BP1si), s773 (DNA-PKcssi), s2085 (BRCA2si), s4173 (DNA2si), s11849 (CtIPsi), s17502 (Exo1si), s8959 (Mre11si), s14952 (Ku80si), s14949 (XRCC4si), s8179 (Lig4si), s21059 (PolQsi1), and s21061 (PolQsi2). Treatment with NU7441 (2 µM; SelleckChem), AZD7648 (2 µM; MedChemExpress), or M3814 (Nedisertib; 2 µM; MedChemExpress) was done either directly before Cas9-sgRNA transfection or 16 h afterward. Following flow cytometry, the different DSB-repair frequencies were quantified using FlowJo software (BD Biosciences). Gating on forward and side-scatter was applied to select the live, single-cell population, followed by gating on either mCherry or iRFP to select transfected cells. In this population careful gating was

applied to quantify the frequencies of BFP$^+$ and GFP$^+$ (DSB-Spectrum V1), BFP$^-$ and GFP$^+$ (DSB-Spectrum V2), or BFP$^-$/mCherry$^-$, BFP$^-$/mCherry$^+$ and GFP$^+$ (DSB-Spectrum V3) cells. These frequencies are plotted directly in Figs. 1d, i, 2d, g, 4d and supplementary figure 4a. For all other experiments, the frequency of each fluorescent subpopulation in the AAVS1sg-transfected cells was subtracted from the frequency of that same population in the BFPsg-transfected cells, and the resulting background-corrected frequencies were normalized to the NTsi-cells.

### TIDE(R)-analysis and SSA-product PCR of the DSB-Spectrum locus
For TIDE(R) analysis and PCR-mediated detection of the SSA repair product, a DSB-Spectrum assay was performed, and genomic DNA isolated using the DNeasy Blood and Tissue kit (Qiagen) according to the manufacturer's instruction. The DSB-Spectrum target region was then PCR amplified using BFP TIDE FWD and BFP TIDE REV primers, or BFP SSA FWD and BFP SSA REV primers. The TIDE PCR product was submitted for Sanger sequencing, either directly from the PCR-reaction mix or after PCR purification using the QiaQuick PCR purification kit (Qiagen) according to the manufacturer's instruction. Sequence files were uploaded for TIDE(R) analysis on (tide.nki.nl). The SSA-PCR product was analyzed by agarose gel electrophoresis and similarly sequenced by Sanger sequencing.

### Western blotting and antibodies
Samples for western blotting were harvested 72–96h after the first round of siRNA transfection. Cells were lysed on ice in RIPA buffer

## Table 1 | List of primers

| Name | Sequence 5' > 3' |
| --- | --- |
| BFP TIDE FWD | CGTAACAACTCCGCCCCATT |
| BFP TIDE REV | GGGTGTTCTGCTGGTAGTGG |
| BFP SSA FWD | CGGTTTGACTCACGGGGATTTC |
| BFP SSA REV | CGGGCCACAACTCCTCATAAAG |
| FANCA SSA FWD | TTACAGTCTGGGCTGCAGTG |
| FANCA SSA REV | AAAGCCCAGAATCAGACGGG |
| BRCA1 SSA FWD | TCAGTGCCTGTTAAGTTGGC |
| BRCA1 SSA REV | CTGGCAACATCTCTTTATTGAGCA |
| BTK SSA FWD | GGGCCAATGACACACAAAGG |
| BTK SSA REV | ACTCTCCCTTCACAGGTGGT |
| SPAST SSA FWD | ACTGGTCCCTTTTGTGAGCAT |
| SPAST SSA REV | TGAGCTGGAACCACATAGTCCT |
| BTK T7 temp FWD 1 | GAGTCAGTCCCTGTTGGGTG |
| BTK T7 temp REV 2 | CCCTATAGTGAGTCGTATTACCG GCCTGCACCTTTTAATATTAAAAG |
| BTK T7 temp FWD 3 | TAATACGACTCACTATAGGGGCA TGGTGGCTCATGC |
| BTK T7 temp REV 4 | CAAGCAGCACTCTCCCTTCA |
| BTK HR FWD | TAATACGACTCACTATAGGG |
| BTK HR REV | ACTGACATGGACAAGCCCTG |
| HBB SSA FWD | AACAGCCAATCTCAGGGCAA |
| HBB SSA REV | CACTGACCTCCCACATTCCC |
| HBB TIDE FWD | GGAGGGCTGAGGGTTTGAAG |
| HBB TIDE REV | CTTTTGTTATACACAATGTTAAGGCAT |
| Splink adapter FWD | phos-CATGGTTGTTAGGACTGGAGGGG AAATCAATCCCCT |
| Splink adapter REV | CCTCCACTACGACTCACTGAAGtGCA AGCAGTCCTAACAACCATGT |
| P-short V2 | CCTCCACTACGACTCACTGAAGTGC |
| Splink N-out | GCGATCTAATTCTCCCCCGC |
| Splink C-out | CGGGACGTCCTTCTGCTAC |
| Splink P-nested | GTGCAAGCAGTCCTAACAACCATG |
| Splink N-in | TTTGGCGTACTCACCAGTCG |
| Splink C-in | TCAATCCAGCGGACCTTCC |
| GAPDH qRT FWD | CTCTGCTCCTCCTGTTCGAC |
| GAPDH qRT REV | ACCAAATCCGTTGACTCCGA |
| PolQ qRT F1 | GAGTGGACACAGTAGGCGAG |
| PolQ qRT R1 | TGCAGCCAAAAATGTGCAGG |
| PolQ qRT F2 | AAGCAAGCTAGCTCACCTCA |
| PolQ qRT R2 | AAGGTGGAACCAAACTGGCA |
| DNA2 qRT F1 | AGCTTTCTGTGTTACCCCCG |
| DNA2 qRT R1 | TCTTCTGAAATAGCTCCGCCG |
| DNA2 qRT F2 | GGTTCCGCTGTCTTTTCTGTC |
| DNA2 qRT R2 | CTTTCTTCTGAAATAGCTCCGC |

(50 mM Tris-HCl pH 8.0, 1 mM EDTA, 1% Triton-X100, 0.5% Sodium Deoxycholate, 0.1% SDS, 150 mM NaCl) supplemented with complete EDTA-free Protease Inhibitor Cocktail tablets (Roche), 2 mM MgCl$_2$ and Benzonase Nuclease (100 U/ml; Merck Millipore). Lysates were cleared by centrifugation at $16,873 \times g$ for 15 min, and protein concentration was determined by a BCA assay (Pierce). Next, SDS-sample buffer (pH 6.8) was added to the lysates, followed by incubation at 95 °C for 5 min. Subsequently, equal amounts of protein for each sample were loaded on 4–15% Criterion TGX precast midi protein gel (Bio-Rad). After SDS-PAGE, proteins were transferred to nitrocellulose membranes using a standard tank electrotransfer protocol. Membranes were blocked using either 5% skim milk or Blocking buffer for fluorescent WB

(Rockland) in PBS, followed by antibody staining in Blocking buffer for fluorescent WB (Rockland) diluted 1:1 in PBS with 0.1% Tween-20. After secondary antibody staining, the membranes were imaged on an Odyssey CLx scanner (LI-COR BioSciences) running ImageStudio ver. 5.2. Image analysis was done using ImageStudio Lite (LI-COR BioSciences). Primary antibodies used were RabbitαBRCA1 (Millipore 07-434; 1:2000), Rabbitα53BP1 (Novus NB100-304; 1:2000), RabbitαKu80 (Santa-Cruz sc-9034; 1:200), MouseαDNA-PKcs (Abcam ab44815, 1:700), MouseαXRCC4 (Signalway antibody 40455, 1:1000), RabbitαLigase-4 (Abcam ab 193353, 1:1000), MouseαTubulin (Sigma T6199, 1:3000), MouseαBetaActin (Sigma, 1:10,000), MouseαBRCA2 (Merck-Millipore OP95, 1:500), MouseαRad52 (Santa-Cruz sc-365341, 1:100,), RabbitαExo1 (Abcam ab95068; 1:2000), MouseαCtIP (Millipore MABE1060; 1:1000), and RabbitαMre11 (kind gift from prof. Roland Kanaar, Erasmus MC, Rotterdam, the Netherlands; 1:3000)[64]. Secondary antibodies used were GoatαMouse or GoatαRabbit labeled with either IRDye 680 or IRDye 800 (LI-COR, 1:10,000).

### RT-qPCR

For qRT-PCR quantification of DNA2 and PolQ knockdown efficiency, RNA was isolated using the RNeasy kit (Qiagen) according to the manufacturer's instructions. Subsequently, RNA was reverse transcribed using the Promega Reverse Transcription Kit according to the manufacturer's instruction. Next, the PCR-reaction mix was assembled using the Promega GoTaq qPCR master mix, and the qPCR was performed on a CFX384 C1000 Touch thermal cycler (BioRad). GAPDH was amplified as a reference transcript. All primers are indicated in Table 1. Data were analyzed using BioRad CFX Manager Software version 3.1. Primer efficiency was determined to be close to 100%. Normalized transcript levels were determined using the ΔΔCt method.

### Clonogenic survival assays

HEK 293T DSB-Spectrum_V2 or U2OS DSB-Spectrum_V2 cell-lines were transfected with pX459-Cas9-sgRNA-iRFP constructs containing either the AAVS1-targeting control sgRNA of BFP-targeting sgRNA 2. Transfections were done with Lipofectamine 2000 (Invitrogen) according to the manufacturer's instructions. At 48 h after transfection, iRFP-positive cells were collected using FACS and plated in six-well plates, either 400 cells per well for U2OS cells, or 800 cells per well for the HEK 293T cells. The HEK 293T cells were plated in Poly-L-lysine coated plates to prevent detachment during later washing and staining steps. Of note, while performing the FACS to collect iRFP-positive cells, analysis of BFP and GFP expression in this population was also performed to determine the frequency of mutagenic repair and HR. Directly after plating, cells were either left untreated or treated with puromycin (2 μg/ml; Invivogen). At 10d after plating, cells were washed with PBS and stained and fixed with a staining solution containing 0.4% Crystal Violet and 20% methanol. Plates were scanned, and colonies were counted using ImageJ.

### SSA and HR at Alu-loci

To study SSA between Alu elements at endogenous loci in the *BTK*, *SPAST*, *FANCA*, and *BRCA1* genes, HEK 293T or HEK 293T DSB-Spectrum_V2 cells were reverse-transfected with siRNA using Lipofectamine RNAiMAX (Invitrogen) according to manufacturer's protocol. At 48 h after siRNA transfection, cells were transfected with Cas9-sgRNA constructs and harvested 48 h later to isolate genomic DNA using the DNeasy Blood and Tissue kit (Qiagen) according to the manufacturer's instruction. Next, the SSA repair product was PCR amplified using GoTaq G2 polymerase (Promega) in 25 μl reaction volumes containing 100–250 ng genomic DNA as a template. When product yield was compared between samples, the exact amount of genomic DNA template was equalized. The primers used for amplification of the SSA repair products are indicated in Table 1. The regular GoTaq G2 cycling parameters were adapted to start with 10 cycles of touchdown PCR, starting at an annealing temperature of 68 °C,

followed by −1 °C per cycle. This was followed by 26 cycles of regular PCR using an annealing temperature of 57 °C. To amplify the SSA repair product from the BFP locus, DMSO was added to the PCR-reaction mixture to an end-concentration of 5%. After PCR, reaction mixtures were directly analyzed by DNA gel electrophoresis.

To study HR at the *BTK* locus, a repair template was generated containing homology arms and a T7 primer binding site at the Cas9-target site by overlap mutagenesis PCR using the BTK T7 Temp primers indicated in Table 1. The resulting 1251 bp product was PCR purified. HR assays were performed essentially as described for SSA assays, but including co-transfection of the T7 repair template. Amplification of the HR repair product was done using the BTK HR primers indicated in Table 1. PCR amplification parameters were as described for amplification of the SSA repair product, but using an annealing temperature of 55 °C during the 26 cycles after the touchdown PCR.

### SSA and HR at the *HBB* locus
SSA at the *HBB* locus was studied as described above for the Alu-loci. However, PCR amplification of the SSA repair product was done using LongAmp Taq DNA polymerase (NEB) according to the manufacturer's protocol. The cycling parameters were adapted to start with five cycles touchdown PCR, starting at an annealing temperature of 64 °C, followed by −1 °C per cycle. This was followed by 25 cycles of regular PCR using an annealing temperature of 59 °C

To study HR-mediated gene conversion between the *HBB* and *HBD* genes, HEK 293T cells were reverse-transfected with siRNA using Lipofectamine RNAiMAX (Invitrogen) according to the manufacturer's protocol. At 24 h after siRNA transfection, cells were transfected with Cas9-sgRNA-GFP constructs and sorted on GFP + cells 48 h later, followed by isolation of genomic DNA using the DNeasy Blood and Tissue kit (Qiagen) according to the manufacturer's instruction. Next, the Cas9-target region in the *HBB* gene was PCR amplified using the HBB TIDE primers indicated in Table and using Q5 High-Fidelity Polymerase (NEB) according to the manufacturer's instructions with an extension time of 45 s. for 30 cycles. Next, the product representing the PCR-amplified target site was extracted from an agarose gel, sequenced, and analyzed by TIDER[31]. To generate a reference chromatogram representing the HBB to HBD converted gene, the HBB target region was amplified and mutated using PCR-mediated mutagenesis to generate, adjacent to the Cas9-target site in the *HBB* gene, a total of six point mutations that distinguish the *HBD* gene from the *HBB* gene in this region. This PCR product was sequenced and uploaded as a reference chromatogram for TIDER analysis. For each experiment, the PCR products were analyzed by sequencing and TIDER three times (technical replicates), and the experiment was performed three times independently (biological replicates).

### Splinkerette PCR
To identify the genomic integration sites of the DSB-Spectrum reporter cassettes, a Splinkerette PCR protocol was performed adapted from Yin et al.[28]. In short, genomic DNA was isolated using the DNeasy Blood and Tissue kit (Qiagen) according to the manufacturer's instructions. Next, 2 µg of genomic DNA was digested with TfiI (NEB) and StyI (NEB), incubating for 6 h at 37 °C and another 6 h at 65 °C. The digested DNA was purified from the restriction mix using the PCR purification kit (Qiagen) according to the manufacturer's instruction, followed by incubation with dNTPs and GoTaq G2 polymerase (Promega) at 72 °C for 30 min to add A-overhangs. Next, the Splink adapter primers were annealed to generate the splinkerette adapter. The adapter was ligated to the A-tailed, digested genomic DNA by O/N incubation at 16 °C with T4 DNA ligase (NEB). Subsequently, the ligated product was purified from the ligation reaction mixture using PCR purification and used as a template in a PCR reaction using the P-Short V2 primer that binds the adapter and either the Splink N-out or the Splink C-out primers,

which bind in the DSB-Spectrum cassette at the 5′-end or 3′-end, respectively. PCR amplification was done using LongAmp Taq DNA polymerase (NEB). Next, 1 µl of this first PCR reaction was used as a template for a second, nested PCR using LongAmp Taq polymerase with the primer Splink P-nested, together with either Splink N-in or Splink C-in primers. After amplification, the PCR-reaction mixture was directly cloned into the TOPO-TA backbone (ThermoFisher Scientific) and transformed into bacteria. Plasmid DNA was extracted from bacterial colonies and sequenced. In some cases, after the second, nested PCR, the reaction mixture was analyzed by DNA gel electrophoresis, and the high abundance PCR products were gel-extracted. The extracted PCR products were subsequently amplified by repeating the second PCR, followed by TOPO-cloning and sequence analysis. All integration sites were verified by PCR.

### Southern blotting
Genomic DNA (gDNA) was isolated using the DNeasy Blood and Tissue kit (Qiagen) according to the manufacturer's instructions. Next, 4 µg of genomic DNA was digested either with *Xba*I (Fermentas) or with *Sap*I (NEB), the latter in combination with *Pac*I (NEB) for the gDNA isolated from U2OS DSB-Spectrum_V2 cells. The digestion mix was incubated overnight at 37 °C. Next, restriction fragments were separated by gel electrophoresis on a 0.8% agarose gel in TAE buffer, running for 16 h at 35 V. The DNA was transferred to a Hybond XL nylon membrane (GE Healthcare) by southern blotting. After neutralization in 2×SSC (300 mM NaCl, 30 mM Sodium Citrate), 0.2 M Tris pH 7.5 buffer, membranes were dried, and DNA was crosslinked to the membrane with a Stratagene UV Stratalinker 1200 (auto crosslink setting). Next, membranes were hybridized (O/N 65 °C) in hybridization buffer (0.125 M NaHPO4 pH 7.2, 10% PEG 6000, 0.25 M NaCl, 1 mM EDTA, 7% SDS, 1% 10 mg/ml fish sperm) with a $^{32}$P-labeled puro probe (600 bp, homologous to the puromycin resistance gene). The membranes were washed at 65 °C once for 10′ in 2 × SSC,0.1% SDS, and twice for 10′ in 1 × SSC, 0.1% SDS. Hybridized blots were exposed for 48 h to a phosphor imager screen and imaged on an Amersham Typhoon (Cytiva). Before reprobing with an mCherry probe (648 bp, homologous to the mCherry gene), membranes were stripped in boiling hot 0.1× SSC buffer, 0.1% SDS. mCherry probe washing conditions after hybridization: once for 10′ in 1 × SSC,0.1% SDS, followed by twice for 10′ in 0.3 × SSC, 0.1% SDS.

### Statistical analysis
All statistical analysis was performed using Graphpad Prism software. In the case of direct statistical comparison between two samples a two-sided Student's *t*-test was performed. In case multiple samples were tested against a control, a one-way ANOVA with post-hoc multiple comparisons was performed.

### Reporting summary
Further information on research design is available in the Nature Research Reporting Summary linked to this article.

## Data availability
The data that support the findings of this study are available from the corresponding authors upon reasonable request. Source data are provided with this paper.

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

## Acknowledgements

We thank Patrick van der Vliet and Dr. Richard Lemmers (LUMC, Leiden, the Netherlands) for their assistance with the southern blotting experiments. We thank the Robert A. Swanson (1969) Biotechnology Center (Koch Institute, MIT, Cambridge MA, USA) and the LUMC Flow cytometry core facility (LUMC, Leiden, the Netherlands) for assistance. This research was supported by National Institute of Health grants R01-ES015339 (M.B.Y.), R35-ES028374 (M.B.Y.), R01-CA226898 (M.B.Y.), by the joint Cancer Research UK and Brain Tumour Charity funded Brain Tumour Award C42454/A28596 (M.B.Y.), by the Charles and Marjorie Holloway Foundation (M.B.Y.), by the MIT Center for Precision Cancer Medicine, by fellowships from the Dutch Cancer Society (BUIT 2015-7546; B.v.d.K.), the Ludwig Center at MIT's Koch Institute (B.v.d.K.), by funding from the Leiden University Medical Center regulation for MSCA-IF Seal of Excellence awardees (B.v.d.K.), and by ERC Consolidator (ERC-CoG-617485) and NWO-VICI grants (VI.C.182.052; H.v.A.). Support was also provided by the Cancer Center Support Grant P30-CA14051 from the National Cancer Institute and the Center for Environmental Health Sciences Support Grant P30-ES002109 from the National Institute of Environmental Health Sciences (M.B.Y.).

## Author contributions

B.v.d.K. and M.B.Y. conceived the project. B.v.d.K., A.K., H.v.A., and M.B.Y. designed the experiments. B.v.d.K. and A.K. performed the experiments. B.v.d.K., A.K., H.v.A., and M.B.Y. analyzed the data. B.v.d.K. and M.B.Y. wrote the paper. A.K. and H.v.A. edited the manuscript. H.v.A. and M.B.Y. supervised the study. B.v.d.K., H.v.A., and M.B.Y. obtained funding for the work.

## Competing interests

The authors declare no competing interests.

## Additional information

¹Department of Human Genetics, Leiden University Medical Center, Einthovenweg 20, 2333 ZC Leiden, the Netherlands. ²Koch Institute for Integrative Cancer Research, MIT Center for Precision Cancer Medicine, Departments of Biology and Bioengineering, Massachusetts Institute of Technology, Cambridge, MA, USA. ³Department of Surgery, Beth Israel Deaconess Medical Center, Divisions of Acute Care Surgery, Trauma, and Critical Care and Surgical Oncology, Harvard Medical School, Boston, MA, USA. ⁴Surgical Oncology Program, National Cancer Institute, National Institutes of Health, Bethesda, MD, USA. ✉e-mail: H.van.Attikum@lumc.nl; myaffe@mit.edu

