## [Peer Review File · Nature Communications]

Multi-Pathway DNA-Repair Reporters Reveal Competition between End-joining, Single-Strand Annealing and Homologous Recombination at Cas9-induced DNA Double-Strand BreaksREVIEWER COMMENTS

Reviewer #1 (Remarks to the Author):

In this manuscript by van de Kooij and colleagues, the authors describe three distinct reporter to monitor repair pathway usage in mammalian cells. These fluorescent reporters are activated by CRISPR/Cas9-based double-strand break formation. DSB-Spectrum_V1 can distinguish error-free canonical non-homologous end-joining (c-NHEJ) from homologous recombination (HR), DSB-Spectrum_V2 can distinguish mutagenic- c-NHEJ repair versus HR and DSB-Spectrum_V3 can distinguish c-NHEJ from HR and single strand annealing (SSA). They have integrated these reporters in HEK 293T and U2OS cells to validate that they were functional. To this end, they treated cells with a small molecule inhibitor for DNAPKcs (NU7441), which results in a decrease in c-NHEJ and an increase HR and SSA usage. Also, they have used siRNA's targeting a variety of key repair pathway proteins, to successfully validate the reporters. Interestingly, using these reporters they find that SSA (a repair pathway that requires end-resection) is used much more frequently than previously thought, leading to large deletions of DNA between homologous sequences. This knowledge about SSA usage with and without inhibition of DNAPKcs is of importance in the context of CRISPR/Cas9-mediated gene editing.

Taken together, the study describes a novel and very useful reporter system which to systematically measure mutagenic-end joining, SSA, HR in the same cell after a DSB created with CRISPR/Cas9 technology. The experiments that were performed to validate these reporters are solid, and the study contains several important and unexpected new insights on the balance of repair pathway activities that compete for repair of a double-stranded break. Also, the results presented in this study shed new light on the possible outcomes of a CRISPR/Cas9-based genome-editing strategy. The reporters presented in this study will be very useful for the DNA Damage field and for those that use CRISPR-editing. I think the manuscript merits publication in Nature Communications, but I do have a number of comments that should be addressed, as well as a number of issues that I feel need clarification before publication is warranted.

Comments:

- Figure 1C: The FACS plot shows that there is a significant percentage of BFP+ cells present when the cells containing the DSG-spectrum-V1 reporter are challenged with the AASV1 gRNA (and possibly even when unchallenged?). Can the authors explain this?
- The FACS plots presented in figure 2C/2G and supplementary figure 2E contain a specific population of HEK 293T and U2OS DSB_Spectrum_V2 cells that appears to be double positive (BFP+ / GFP+). Is the gating incorrect? Or do the authors have an explanation for this.
- Figure 3B: The indels measured by TIDER show that there are high amounts of -12/-11 deletions. These are most likely produced through alternative end-joining (a-EJ). It would be an interesting addition to the paper if the authors could show that this is definitely a-EJ by using specific inhibitors or show sequence similarities. For example, show that the inhibition of PolQ leads to a decrease of -12/-11 indels.
- Figure 4 B2 and Figure 1 & 2: in DSB-spectrum_V1 and DSB-spectrum_V2 the GFP used for HR is indicated in the figures as GFP. In the last DSB-spectrum reporter, _V3, it is indicated as iGFP. It is not clear what the differences are between the GFP in _V1 and _V2 compared to _V3.
- Figure 5E: In the FA67 Alu Element PCR, a band is observed in the condition transfected with a AAVS1 gRNA. What would be the explanation for this? Is it a possibility to sequence those bands to see if this is a background band, or is this is a naturally occurring SSA-event in those cells?
- The authors use a DNAPKc inhibitor (NU7441) in their study. Recently more potent DNAPKc inhibitors have been reported (AZD7648, Fok et al., 2019 or M3814, Zenke et al., 2020). The authors claim in this study that DNAPKc inhibition leads to an increase of the SSA pathway. It would be of additive value to show this with another inhibitor. It is important to exclude that the results observed in this study are not inhibitor-specific.
- Throughout the paper, cells were transfected with plasmids containing the Cas9, gRNA and a specific fluorophore, to target the various reporters. I had a hard time to understand the combination of

Cas9_gRNA_fluorophore-plasmid with the DSB_Spectrum variants. A schematic representation of the setup of the experiment would therefore be informative for the reader.

- HR and SSA both need extensive resection, but while SSA can occur on a single chromatid, HR requires the presence of a sister chromatid or a chromatid from the homologous chromosome. While resection occurs more readily in S/G2, it is not limited to these cell cycle stages. It would be interesting to measure the relative pathway frequency in non-cycling (i.e. G1 arrested) versus cycling (S/G2) cells. Is there a way to synchronize these cultures? Such an experiment may require shorter incubation times (in the current setting repair is determined 72 hr after induction of the break). Shorter incubation could also shed some light on the relative timing a repair through each pathway.

- In the DSB_Spectrum_V2 and DSB_Spectrum_V3 reporter, restoration of BFP+ could also be achieved by HR in S/G2, where an uncut copy of the BFP gene on the sister chromatid can be used as the template (instead of the GFP). Thus, simply scoring the percentage of GFP+ cells as read-out for the frequency of HR-usage may potentially give rise to an underestimate. This might be difficult to resolve but could the actual frequency of HR be higher? In addition to this, recutting events could also affect the measurement of HR-usage. To address this possibility the authors could harvest an earlier timepoint after transfection of Cas9&gRNA.

Reviewer #2 (Remarks to the Author):

This m/s describes three new reporters of DSB repair that are intended to streamline the measurement of HR, error-free NHEJ, error-prone end joining and SSA in the repair of a site-specific DSB. The reporters have substantial defects that will make them less useful than existing reporters.

First, some repair products are inferred indirectly, and lack a positive identifying fluorescent outcome. Most notable of these is SSA, which is scored only by the loss of an mCherry marker. Hence, the specific ability of SSA to perform homology-based annealing is not measured. As a result, non-SSA deletions will be falsely scored as SSA products by this reporter. This excludes the "SSA reporter" described here as a useful tool. An interesting failure of the authors' SSA reporter is shown in the effect of BRCA1 siRNA. Previous work has shown a substantial defect in SSA in BRCA1-depleted cells. In the current reporter system (Figure 4G), BRCA1 depletion has very modest effects on the frequencies of BFP- mCherry- outcomes. This is a strong indication that the "SSA reporter" may not be specific to SSA events in all cases. The mutagenic end joining outcome, scored as loss of BFP+, suffers from a similar defect. These defects are evident in the high background of BFP- cells in the control sgRNA transfected populations of cells.

Second, the FACS images clearly show that the BFP+ and BFP- populations overlap one another, as do the mCherry+ and mCherry- populations. This defect will limit the ability of these reporters to detect subtle differences between treatment groups.

The biology reported in this m/s largely reproduces previous findings. These findings are used to validate the function of the new reporters. (However, the authors fail to recognize that their "SSA reporter" underreports the impact of BRCA1 depletion on SSA, as discussed above).

One additional technical concern is the method used to test whether each randomly-integrated reporter has been incorporated as a single copy in the reporter clones tested. Southern Blotting is the gold standard method required to validate the structure and copy number of the chromosomally integrated reporter. The method used in this paper is inferior, since the PCR-based test used could fail by chance to detect a second reporter copy or, alternatively, concatemers of reporters integrated in tandem.

Given the evident defects in these reporters, this reviewer advises against publication in Nature Communications.

Reviewer #3 (Remarks to the Author):

In the manuscript "Multi-Pathway DNA Double-Strand Break Repair Reporters Reveal Extensive Cross-Talk Between End-Joining, Single Strand Annealing, and Homologous Recombination" Kooij et al report the development of fluorescent Cas9-based reporter assay systems to monitor multiple DSB repair pathways simultaneously and to investigate crosstalk between them. The authors found that the inhibition of cNHEJ factors can increase the activity of SSA and HR and that the repair event by SSA is more common than generally thought. Importantly short-range resection by CtIP or Mre11 promoted both SSA and HR, while long-range resection by DNA2 or EXO1 promoted SSA and inhibited HR. This is the novel and key finding in this study. Currently the mechanism distinguishing SSA and HR is largely unknown. As outlined in more detail below, additional data is needed to support this finding. The assay systems developed here will be useful to investigate DSB repair pathway crosstalk and to understand outcomes following perturbation of the DSB signaling and repair network. Prior to publication, the following concerns should be addressed.

1. The observation that resection factors decide a DSB pathway, HR or SSA is the key finding of this paper. In the DSB-Spectrum _V3 cells, CtIP and Mre11 promoted SSA and HR. However, DNA2 and EXO1 promoted SSA but inhibited HR (Fig. 6). The authors should test if the similar effect can be detected in the SSA assay between Alu elements using the endogenous genomic loci shown in Fig 5, to confirm it in an additional context.
2. A-EJ (mutagenic EJ) was increased after the depletion of 53BP1 and DNA-PKcs in previous studies, Mol. Cell. 2016; 63:662–673 and NAR Cancer 2020; zcaa017, respectively. By contrast the authors report that mutagenic EJ was reduced by 53BP1si or NU7441 in Fig. 4. It is probably because mutagenic EJ detected by DSB-Spectrum _V2 and 3 was mediated by error prone cNHEJ. The authors should consider and discuss this possibility. BFP+ cells include samples repaired by cNHEJ. BFP- cells can be generated either by cNHEJ or by a-EJ in Fig. 2, 4, 5, 6.
3. The reporter sequence was stably integrated into human cell lines, but Cas9 and gRNA were transfected every time to monitor DSB repair. The transfection efficiency must influence the result outcome. How did the authors control for transfection efficiency in the experiments?
4. +1 T insertion at CRISPR-induced DSBs was the most abundant among all Indel events (Fig. 3A). It has been reported that Cas9-catalyzed DNA cleavage produces 1 bp staggered ends rather than blunt ends (Sci. Rep. 2016; 5:37584 & NAR Cancer 2020; zcaa017). This is a preferred substrate of c-NHEJ. This could explain the reason why NU7441 reduced the +1 T insertion.
5. Knock down of endogenous protein level of POLQ and DNA2 after siRNA were not confirmed by immunoblotting in Sup Fig 1E and Fig 6B. Reduction of DNA2 mRNA was shown in Fig 6 C, such experiment was not performed for POLQ. Proper validation of knock-downs should be demonstrated by immunoblotting or qPCR.
6. Mutagenic EJ was detected with DSB-Spectrum _V2 and 3. However, the systems cannot distinguish the types of end-joining, cNHEJ and a-EJ. cNHEJ and a-EJ can both perform mutagenic EJ. Sequencing of DSB repair junctions is needed for further clarification. Indeed, TIDER analysis (sequencing analysis) identified products possibly repaired by a-EJ (Fig. 3B). The authors should reconsider the sentence in introduction "--- DSB-repair phenotypes that can easily be missed by commonly used sequencing approaches that analyze the DSB-repair junction, "(Page 4, line 135-137).
7. The authors should reconsider the sentence "As an alternative to c-NHEJ, DSBs can be repaired by Homologous Recombination (HR) (Page 3, Line 71-72)". It is well established that the repair pathways

are cell cycle dependent and affected by sequence of damage site. HR is a dominant pathway to repair DSBs especially in S/G2 cells.

8. Page 17 Line 524, DBS should be DSB.

Reviewer response letter

Manuscript: NCOMMS-21-05354A

Title: *Multi-Pathway DNA Double-Strand Break Repair Reporters Reveal Extensive Cross-Talk Between End-Joining, Single Strand Annealing, and Homologous Recombination*

Authors: *Bert van de Kooij, Alex Kruswick, Haico van Attikum, and Michael B. Yaffe*

We would like to thank the reviewers for their careful evaluation of our manuscript. We have addressed their comments and suggestions by performing numerous additional experiments, which has resulted in considerable changes to the manuscript and the inclusion of multiple new figure panels. We believe this has further solidified the experimental basis of our conclusions and has overall improved the manuscript. Please find a point-by-point reply to all reviewer comments and suggestions below.

REVIEWER COMMENTS

Reviewer #1 (Remarks to the Author):

In this manuscript by van de Kooij and colleagues, the authors describe three distinct reporter to monitor repair pathway usage in mammalian cells. These fluorescent reporters are activated by CRISPR/Cas9-based double-strand break formation. DSB-Spectrum_V1 can distinguish error-free canonical non-homologous end-joining (c-NHEJ) from homologous recombination (HR), DSB-Spectrum_V2 can distinguish mutagenic- c-NHEJ repair versus HR and DSB-Spectrum_V3 can distinguish c-NHEJ from HR and single strand annealing (SSA). They have integrated these reporters in HEK 293T and U2OS cells to validate that they were functional. To this end, they treated cells with a small molecule inhibitor for DNAPKcs (NU7441), which results in a decrease in c-NHEJ and an increase HR and SSA usage. Also, they have used siRNA's targeting a variety of key repair pathway proteins, to successfully validate the reporters. Interestingly, using these reporters they find that SSA (a repair pathway that requires end-resection) is used much more frequently than previously thought, leading to large deletions of DNA between homologous sequences. This knowledge about SSA usage with and without inhibition of DNAPKcs is of importance in the context of CRISPR/Cas9-mediated gene editing.

Taken together, the study describes a novel and very useful reporter system which to systematically measure mutagenic-end joining, SSA, HR in the same cell after a DSB created with CRISPR/Cas9 technology. The experiments that were performed to validate these reporters are solid, and the study contains several important and unexpected new insights on the balance of repair pathway activities that compete for repair of a double-stranded break. Also, the results presented in this study shed new light on the possible outcomes of a CRISPR/Cas9-based genome-editing strategy. The reporters presented in this study will be very useful for the DNA Damage field and for those that use CRISPR-editing. I think the manuscript merits publication in Nature Communications, but I do have a number of comments that should be addressed, as well as a number of issues that I feel need clarification before publication is warranted.

Comments:

Thank you for this very positive evaluation of our manuscript.

1. Figure 1C: The FACS plot shows that there is a significant percentage of BFP+ cells present when the cells containing the DSB-spectrum-V1 reporter are challenged with the AASV1 gRNA (and possibly even when unchallenged?). Can the authors explain this?

Reply: Throughout our experiments we indeed find between 0.5% and 4% BFP+ cells in our AAVS1sg control population (see Fig. 1D). The chance that a spontaneous mutagenesis event would cause expression of BFP from the DSB-Spectrum reporter is extremely low, since this requires a very exact deletion of the spacer sequence. It is therefore more likely that the BFP+ cells in the AAVS1sg population are caused by a small population of cells having relatively high levels of background fluorescence. It is well described that cell-endogenous metabolites like NADH can cause background fluorescence, especially in the channels used to detect BFP and GFP (see for example Monici M., *Biotechnol Annu Rev.* 2005;11:227-56. doi: 10.1016/S1387-2656(05)11007-2.). Notably, we introduce Cas9-sgRNA-iRFP(670) constructs into the cells by transfection, and find that the BFP+ cells in the AAVS1sg population are those cells that have very high levels of iRFP(670). We show a flow cytometry plot indicating this for the reviewer's appreciation below (Reviewer Fig. 1A). Stress induced by very high Cas9 and iRFP(670) expression could result in changed levels of metabolites causing background fluorescence. Throughout the manuscript we analyzed all iRFP(670)+ cells and corrected for BFP+ cells in the AAVS1sg population by subtracting them from the % of BFP+ cells in the BFPsg population. Alternatively, researchers using DSB+Spectrum_V1 could simply gate on iRFP(670)-dim cells to get rid of these cells with high background fluorescence (Reviewer Fig. 1B).

Reviewer Figure 1

Reviewer Figure 1. Background BFP expression is mainly detected in iRFP(670)-high cells. (A) Flow cytometry plot of DSB-Spectrum_V1 cells, transfected with a Cas9-AAVS1 sgRNA construct, showing the BFP and iRFP(670) intensities. Red box indicates the BFP+ cells in the iRFP(670)^{high} population (B) Flow cytometry plots of DSB-Spectrum_V1 cells showing the gating on either all iRFP(670)⁺ cells (top panels) or the iRFP(670)^{dim} cells (bottom panels).

2. The FACS plots presented in figure 2C/2G and supplementary figure 2E contain a specific population of HEK 293T and U2OS DSB_Spectrum_V2 cells that appears to be double positive (BFP+ / GFP+). Is the gating incorrect? Or do the authors have an explanation for this.

Reply: The degradation of BFP as well as the synthesis and maturation of GFP does not occur instantaneously after BFP to GFP gene conversion, but is a gradual process. Thus, a fraction of cells will be in an intermediate stage in which they still contain sufficient levels of BFP protein to be BFP positive, but already synthesized enough GFP protein to be GFP positive. To clarify this, the revised manuscript contains a new figure panel showing the kinetics of BFP loss and GFP gain after DSB generation (Fig. S4A). This analysis clearly shows that the GFP positive cells are still partially in the BFP positive gate at 48h, but mostly migrated to the BFP negative gate at 72h and 96h (Fig. S4A, which we now describe in lines 240-247 on page 9 of the revised manuscript). The double positive cells that still remain at later time points could either be cells in which gene conversion occurred at a late time point, or cells that have high background fluorescence, as explained in the reply to comment 1.

3. Figure 3B: The indels measured by TIDER show that there are high amounts of -12/-11 deletions. These are most likely produced through alternative end-joining (a-EJ). It would be an interesting addition to the paper if the authors could show that this is definitely a-EJ by using specific inhibitors or show sequence similarities. For example, show that the inhibition of PolQ leads to a decrease of -12/-11 indels.

Reply: We thank the reviewer for this excellent suggestion. We have done the suggested experiment and performed knockdown of PolQ in DSB-Spectrum_V2 cells, either alone or in combination with inhibition of DNA-PKcs, followed by TIDER analysis. The results of this experiment are shown in new figure 3A, and the new supplementary figure S5A. As the reviewer surmised, the generation of the -12 deletion product was dependent on PolQ expression, consistent with it being the result of repair by a-EJ. We also observed a -9 deletion product specifically in the NU7441-treated cells that showed similar behavior following PolQ depletion. This is now described in lines 300-324 on pages 10-11 of the revised manuscript. Although the initial experiments indicated the presence, albeit at low frequency, of a -11 deletion product in the NU7441-treated samples (old figure 3B), we did not detect this product in the follow-up TIDER experiment depicted in new figure 3A. This is likely a consequence of its low abundance and the somewhat stochastic nature of mutagenic repair.

4. Figure 4 B2 and Figure 1 & 2: in DSB-spectrum_V1 and DSB-spectrum_V2 the GFP used for HR is indicated in the figures as GFP. In the last DSB-spectrum reporter, _V3, it is indicated as iGFP. It is not clear what the differences are between the GFP in _V1 and _V2 compared to _V3.

Reply: There is no difference between the GFP cassettes in the different DSB-Spectrum variants. The term iGFP was coined by Maria Jasin to describe the truncated GFP in her DR-GFP construct. We initially used this nomenclature, but simplified it to just GFP and had apparently not changed it throughout the manuscript. We have now corrected this and changed the iGFP into GFP in figure 4.

5. Figure 5E: In the FA67 Alu Element PCR, a band is observed in the condition transfected with a AAVS1 gRNA. What would be the explanation for this? Is it a possibility to sequence those bands to see if this is a background band, or is this is a naturally occurring SSA-event in those cells?

Reply: As requested by the reviewer, we have now extracted this band from agarose gel and used it as a template in a second PCR reaction to further amplify it. Analysis of the resulting PCR product by agarose gel electrophoresis suggests that it is somewhat smaller than the SSA-repair product amplified from the FA67sg-transfected cells (called FANCAsg in the revised manuscript; Reviewer Fig. 2A). Nevertheless, we TOPO-cloned this PCR product and extracted it from ten colonies, followed by sequencing. We found that 8 of the 10 sequences did not align to the FANCA SSA repair product but in fact aligned to another region in the human genome (Reviewer Fig. 2B). The remaining two sequences did align to the FANCA SSA repair product. We attribute this to likely cross-contamination from the adjacent lane.

Reviewer Figure 2

Reviewer Figure 2. PCR product using FA67 primers in AAVS1-sg transfected cells is mostly a PCR artefact. (A) Genomic DNA obtained in the experiments described in figure 5E of the original manuscript was used as a template for PCR amplification using the FANCA SSA primers. The observed PCR products were gel-extracted and further amplified in a second PCR reaction using the same primers. Picture shows agarose-gel analysis of the products from these second PCR reactions. **(B)** Table showing the result of sequence analysis of TOPO-cloned PCR product from the AAVS1sg cells shown in panel (A).

6. The authors use a DNAPKc inhibitor (NU7441) in their study. Recently more potent DNAPKc inhibitors have been reported (AZD7648, Fok et al., 2019 or M3814, Zenke et al., 2020). The authors claim in this study that DNAPKc inhibition leads to an increase of the SSA pathway. It would be of additive value to show this with another inhibitor. It is important to exclude that the results observed in this study are not inhibitor-specific.

Reply: We thank the reviewer for this excellent suggestion. We have now used the suggested DNA-PKcs inhibitors in a DSB-Spectrum_V3 assay, and found that they caused an increase in SSA and HR, very similar to what we found with NU7441. These results are presented as new figure S7A in the revised manuscript. We conclude that on-target inhibition of DNA-PKcs, irrespective of the inhibitor used, increases SSA and HR.

7. Throughout the paper, cells were transfected with plasmids containing the Cas9, gRNA and a specific fluorophore, to target the various reporters. I had a hard time to understand the combination of Cas9_gRNA_fluorophore-plasmid with the DSB_Spectrum variants. A schematic representation of the setup of the experiment would therefore be informative for the reader.

Reply: We thank the reviewer for this suggestion, and we now have included such a schematic as new figure panel S3A.

8. HR and SSA both need extensive resection, but while SSA can occur on a single chromatid, HR requires the presence of a sister chromatid or a chromatid from the homologous chromosome. While resection occurs more readily in S/G2, it is not limited to these cell cycle stages. It would be interesting to measure the relative pathway frequency in non-cycling (i.e. G1 arrested) versus cycling (S/G2) cells. Is there a way to synchronize these cultures? Such an experiment may require shorter incubation times (in the current setting repair is determined 72 hr after induction of the break). Shorter incubation could also shed some light on the relative timing a repair through each pathway.

Reply: We agree with the reviewer that it would be interesting to study whether SSA can occur in G1. Unfortunately, this experiment is challenging due to the lag time between gene editing and the resulting changes in fluorescence. Cell cycle specific DSB induction, repair and detection therefore require prolonged cell-cycle arrest. Nevertheless, to try and address this issue we aimed to arrest cells in G1 using serum starvation, which is preferred over the use of CDK4/6 inhibitors since HEK 293T cells are refractory to synchronization with these compounds (see Trotter EW, Hagan IM. *Open Biol.* 2020 Oct;10(10):200200. doi: 10.1098/rsob.200200.). First, we synchronized cells in G1 and S-phase with 2.5 mM thymidine (Reviewer Fig. 3), which can be used transiently to achieve synchronization but not throughout the DSB-repair studies because it blocks DNA synthesis. When these cells were released for 24h into medium containing 10% serum they restarted cycling, as expected, and a cell cycle profile comparable to asynchronous cells was observed (Reviewer Fig. 3, panel 3). In addition, we released a second cell population into serum-containing medium for 8h to allow passage through S-phase, and then replaced the medium with serum-free medium, followed by incubation for another 16h. We expected this population to be highly enriched for G1-phase cells, as the lack of serum should prevent entry into the next S-phase. However, no such enrichment of G1-phase cells was observed (Reviewer Fig. 3, panel 4). We therefore conclude that HEK 293T cells cannot be readily G1-arrested by serum-starvation. These technical limitations have prevented us from experimentally investigating cell-cycle stage specific repair in the current manuscript. Given these technical challenges, studying cell-cycle stage specific repair, although interesting, is likely to be a project in itself, and we believe these studies are outside the scope of the current manuscript. We will certainly plan to do such experiments in the future.

With regards to the shorter incubation times, as suggested by the reviewer, we have now included a new figure panel with the repair kinetics of DSB-Spectrum_V2, which is described in more detail in the reply to comment 2 of the reviewer (Fig. S4A).

Reviewer Figure 3

Reviewer Figure 3. Serum starvation does not readily synchronize HEK293T cells in G1. HEK 293T cells were either left growing asynchronously (panel 1), or treated with Thymidine for 20h (panel 2). Subsequently cells were released in thymidine-free medium for 8h. This was followed by 16h incubation in either medium with 10% serum (panel 3), or in serum-free medium. Cells were fixed and stained with DAPI, followed by flow cytometric analysis.

9. In the DSB_Spectrum_V2 and DSB_Spectrum_V3 reporter, restoration of BFP+ could also be achieved by HR in S/G2, where an uncut copy of the BFP gene on the sister chromatid can be used as the template (instead of the GFP). Thus, simply scoring the percentage of GFP+ cells as read-out for the frequency of HR-usage may potentially give rise to an underestimate. This might be difficult to resolve but could the actual frequency of HR be higher? In addition to this, recutting events could also affect the measurement of HR-usage. To address this possibility the authors could harvest an earlier timepoint after transfection of Cas9&gRNA.

Reply: The reviewer correctly points out that any DSB-repair event that restores the original sequence will not be detected, including HR using the BFP gene on the sister chromatid as a template. However, as also pointed out by the reviewer, perfect restoration of the original sequence will allow re-cutting of the target site. These two issues could certainly result in underestimation of repair pathway usage, but this problem is shared by all published reporter systems (summarized in: van de Kooij B, van Attikum H. *Front Genet.* 2022 Feb 14;12:809832. doi: 10.3389/fgene.2021.809832) and should be taken into consideration when analyzing reporter data. For example, in Maria Jasin's classic DR-GFP, in which both GFP alleles are non-functional, HR that restores the original non-functional allele would also not be reported as a GFP-positive HR-event. Nevertheless, despite the shortcomings, reporter systems have proven to be very powerful tools to identify how cell perturbations affect DSB-repair, and we hope that the DSB-Spectrum reporters will be a very important addition to this toolkit.

Reviewer #2 (Remarks to the Author):

This m/s describes three new reporters of DSB repair that are intended to streamline the measurement of HR, error-free NHEJ, error-prone end joining and SSA in the repair of a site-specific DSB. The reporters have substantial defects that will make them less useful than existing reporters.

1. First, some repair products are inferred indirectly, and lack a positive identifying fluorescent outcome. Most notable of these is SSA, which is scored only by the loss of an mCherry marker. Hence, the specific ability of SSA to perform homology-based annealing is not measured. As a result, non-SSA deletions will be falsely scored as SSA products by this reporter. This excludes the “SSA reporter” described here as a useful tool.

Reply: We respectfully disagree. The reviewer argues that non-SSA deletions might occur in DSB-Spectrum_V3 that could be falsely scored as SSA events, and disqualifies DSB-Spectrum as an SSA reporter on this sole assumption. The results presented in our manuscript demonstrate that there is no basis for this concern.

First, the reviewer states that SSA is “scored only by the loss of the mCherry marker”. This is factually incorrect. SSA is detected by the combination of loss of BFP and mCherry.

Second, we directly validate that DSB-Spectrum_V3 reports on SSA by knockdown of well-described pro-SSA factors like Rad52 and end-resection factors (Mre11, CtIP, Exo1 and DNA2), and knockdown of the anti-SSA factor BRCA2. These studies showed that loss of Rad52 and the end-resection factors resulted in decreased BFP and mCherry double negative cells (Fig. 4E), while loss of BRCA2 resulted in increased BFP and mCherry double negative cells (Fig. 6A), confirming the utility of DSB-Spectrum_V3 as an SSA-reporter.

Finally, we show that loss of both BFP and mCherry is the direct result of SSA by sequencing the repair product itself. These data indicate that the majority of combined BFP and mCherry loss is the result of a specific deletion event caused by annealing of the homologous BFP and GFP sequences. This is demonstrated by PCR amplification across the complete DSB-Spectrum locus, which generated a distinct ~1300 bp repair product that we confirmed by sequence analysis to be the result of the predicted SSA repair event (Fig. 3D, 5C, 5F, S6A of the revised manuscript). Furthermore, knockdown of Rad52 decreased the yield of this specific SSA repair product, and inhibition of DNA-PKcs increased the yield of this product, with effect sizes similar to those measured by flow cytometric quantification of the BFP⁻, mCherry⁻ population (Compare Fig. 5C, 5F, to Fig. 4D, 4E, of the revised manuscript). Given the similarity in results obtained by flow cytometry and PCR analysis of the specific SSA repair product, we can only conclude that the contribution of non-SSA deletions events to the population that is BFP and mCherry negative is negligible.

Nevertheless, even if DSB-Spectrum_V3 would falsely attribute a small minority of deletion events to SSA, this would not directly disqualify it as an SSA-reporter. A reporter like EJ2-GFP, for example, has proven to be a valuable tool in DSB-repair research, despite the fact that a minority (~10%) of the GFP⁺ repair products are not caused by the repair pathway it was designed to detect (MMEJ) (Bennardo N, et al., *PLoS Genet.* 2008 Jun 27;4(6):e1000110. doi: 10.1371/journal.pgen.1000110.). This demonstrates that a reporter can be a useful tool even if even if a minority of repair events are scored as false positives.

Furthermore, the reviewer appears to suggest that SSA-reporters with a positive identifying fluorescent outcome are preferred over loss-of-fluorescence reporters like DSB-Spectrum. We strongly disagree with this notion. Detection of a very specific repair product has the disadvantage of scoring false negatives, *i.e.* SSA events that are missed by the reporter. This is exemplified by our sequence analysis of the SSA repair products derived from DSB-Spectrum, which showed that some SSA products contain small deletions (Fig. S6A, SSA seq 2). Such an SSA event would be missed if the detection of SSA depended on restoration of a fluorescent protein encoding gene. Thus, a well-validated loss-of-fluorescence reporter like DSB-Spectrum will actually provide a more accurate quantification of the total amount of possible SSA-events than a gain-of-fluorescence reporter that only detects a very specific genetic outcome.

2. An interesting failure of the authors' SSA reporter is shown in the effect of BRCA1 siRNA. Previous work has shown a substantial defect in SSA in BRCA1-depleted cells. In the current reporter system (Figure 4G), BRCA1 depletion has very modest effects on the frequencies of BFP- mCherry-outcomes. This is a strong indication that the "SSA reporter" may not be specific to SSA events in all cases.

Reply:

The reviewer points out that the observed SSA-defect in BRCA1-depleted cells is relatively modest in our experiments compared to previous work. We regret that the reviewer does not cite any of this previous work, which would have allowed us to perform a detailed comparison between our experiments and the data that he/she refers to. Furthermore, the reviewer interprets this discrepancy as an indication that DSB-Spectrum_V3 might not always specifically detect SSA events. This reviewer comment is actually a follow-up on comment 1, so we would like to point to our reply to comment 1 for a summary of the experimental evidence that demonstrates the validity of DSB-Spectrum_V3 as an SSA-reporter.

Nevertheless, to directly address the reviewer's concern about this specific experiment, we depleted BRCA1 by RNAi in DSB-Spectrum cells, and measured SSA by PCR analysis of the specific and sequence-verified SSA repair product, rather than by flow cytometry. As shown in reviewer figure 4, the yield of this specific SSA repair product was decreased by ~24% in BRCA1 knockdown cells compared to control cells. This is highly comparable to the ~20% reduction that we observed by flow cytometric quantification of the BFP-, Cherry- population in DSB-Spectrum_V3 experiments (Fig. 4G). Thus, DSB-Spectrum_V3 reliably quantifies SSA repair in these BRCA1 knockdown experiments, as in all other experiments (see reply to comment 1).

Importantly, we would also like to point out that the strength of the SSA-defect in BRCA1-depleted cells highly varies across the reports that published this phenotype. Using the reporter SA-GFP in mouse ES cells, it was reported that SSA in BRCA1^{-/-} cells was reduced to, on average, about 55% of the frequency of SSA in WT cells (Stark JM, et al., *Mol Cell Biol.* 2004 Nov;24(21):9305-16. doi: 10.1128/MCB.24.21.9305-9316.2004.). However, the standard deviations are large, suggesting high variability between experimental repeats. In a second published report, RNAi-mediated depletion of BRCA1 in U2OS cells carrying the SA-GFP reporter reduced SSA to ~20% of the frequency of SSA in control cells (Anantha RW, et al., *Elife.* 2017 Apr 11;6:e21350. doi: 10.7554/eLife.21350.). Our data and these published data together indicate that the strength of the SSA-phenotype observed upon BRCA1-depletion is

highly dependent on the experimental context, *i.e.* on the cell-line used, on the method of BRCA1 depletion and on the level of BRCA1 depletion.

Reviewer Figure 4

Reviewer Figure 4. BRCA1 depletion causes a modest SSA-defect in HEK 293T DSB-Spectrum cells. HEK 293T DSB-Spectrum_V2 cells were transfected with a non-targeting control siRNA or a BRCA1-targeting siRNA, and subsequently re-transfected to express an AAVS1-targeting control sgRNA or a BFP sgRNA targeting the DSB-Spectrum locus. Next, the target locus in DSB-Spectrum was PCR amplified with primers designed to detect the specific SSA repair product.

3. The mutagenic end joining outcome, scored as loss of BFP+, suffers from a similar defect. These defects are evident in the high background of BFP- cells in the control sgRNA transfected populations of cells. Second, the FACS images clearly show that the BFP+ and BFP- populations overlap one another, as do the mCherry+ and mCherry- populations. This defect will limit the ability of these reporters to detect subtle differences between treatment groups.

Reply: The reviewer first states that the loss of BFP as read-out for mutagenic end-joining has a similar defect as the SSA readout, *i.e.* it measures loss of expression which could be caused by other events than mutagenic end-joining. We respectfully disagree with the reviewer, based on similar arguments as we outlined above for SSA-repair including direct sequence analysis and genetic and pharmaceutical manipulation of NHEJ factors. We performed sequence analysis to examine which repair events cause BFP loss, and find that c-NHEJ, together with SSA, and a minor fraction of MMEJ, are responsible for loss of BFP expression using DSB-Spectrum_V2 (Figs. 2 and 3). This is why we consistently state that BFP loss in DSB-Spectrum_V2 is the result of mutagenic repair without claiming that it is entirely through mutagenic end-joining, since SSA could also be involved. To distinguish the SSA population from the mutagenic end-joining population we generated DSB-Spectrum_V3. Using DSB-Spectrum_V3 cells, we find that inhibition of c-NHEJ with three different inhibitors of DNA-PKcs strongly reduces the frequency of BFP negative cells. These data together are consistent with loss of BFP in DSB-Spectrum_V3 cells being the result of mutagenic end-joining (Fig. 4D and Fig. S7A of the revised manuscript).

In her/his second statement, the reviewer correctly points out that about 4% of the cells in the control sgRNA population are gated as BFP negative cells. This is explained by the gating procedure, in which we set the gate close to the BFP positive population, to ensure that we include all BFP negative cells. Due to the normal distribution of BFP intensity per cell across the population, tight gating on BFP negative cells can result in inclusion of a small fraction of BFP-positive cells that are at the edge of the distribution. We correct for this inclusion of BFP-dim cells by subtracting the BFP negative population in the AAVS1sg cells from the BFP positive population in the BFPsg cells.

The reviewer next mentions that there is an overlap between the positive and negative populations when analyzing BFP and mCherry expression. Notably, it is important to point out that the separation between these populations is very dependent on the analysis time after Cas9 transfection, because the loss of BFP and mCherry protein is gradual as it requires

degradation of the pre-existing fluorescent protein pool. To more clearly document this, we performed additional experiments, that are now included in the revised manuscript (Fig. S4A, S6B) showing a time course analysis of Cas9-transfected DSB-Spectrum_V2 and _V3 cells. These experiments clearly show that separation between the BFP+/mCherry+ and BFP-/mCherry- populations is very good at 72h, the time point we used throughout our experiments (Fig. S4A, S6B). Even further separation between the populations can be obtained by using a 96h incubation time, and future users of our reporter systems could therefore decide to implement these longer incubation times if wanted. These results are now described in lines 240-247 on page 9, and in lines 369-375 on pages 12-13 of the revised manuscript.

Finally, the reviewer claims that the DSB-Spectrum reporters might be limited in their ability to detect subtle differences between treatments groups, based on the minor overlap that exists between the positive and negative BFP/mCherry populations. We respectfully disagree with this comment. Targeting Cas9 to DSB-Spectrum results in 50%-80% BFP- cells, and ~25% mCherry- cells, depending on the time point of read-out (Fig. S4A, S6B), which is a very substantial increase from the ~5% of these populations in the control cells. Thus, there is a very large window to detect subtle differences between treatment groups.

4. The biology reported in this m/s largely reproduces previous findings. These findings are used to validate the function of the new reporters. (However, the authors fail to recognize that their “SSA reporter” underreports the impact of BRCA1 depletion on SSA, as discussed above).

Reply: We do not completely understand what the underlying concern or suggestion is that the reviewer intends to make with this comment. Indeed, we reproduced previous findings to specifically validate the read-outs in these new reporter systems. If the reviewer intends to suggest that there is not sufficient novelty presented in our manuscript, then we would like to point out that we describe three novel reporter systems, reveal that SSA contributes more to DSB-repair than generally considered, and show that Exo1/DNA2 mediated long-range end-resection is not required for HR, and even inhibits it at the DSB-Spectrum locus. These novel findings are clearly appreciated by the other reviewers, as can be taken from their reports. Reviewer #1 for example states: *“Interestingly, using these reporters they find that SSA (a repair pathway that requires end-resection) is used much more frequently than previously thought, leading to large deletions of DNA between homologous sequences. This knowledge about SSA usage with and without inhibition of DNAPKcs is of importance in the context of CRISPR/Cas9-mediated gene editing. Taken together, the study describes a novel and very useful reporter system which to systematically measure mutagenic-end joining, SSA, HR in the same cell after a DSB created with CRISPR/Cas9 technology.”* Similarly, reviewer #3 states: *“Importantly short-range resection by CtIP or Mre11 promoted both SSA and HR, while long-range resection by DNA2 or EXO1 promoted SSA and inhibited HR. This is the novel and key finding in this study.”* We have already addressed the BRCA1 depletion phenotype in our reply to comment 2.

5. One additional technical concern is the method used to test whether each randomly-integrated reporter has been incorporated as a single copy in the reporter clones tested. Southern Blotting is the gold standard method required to validate the structure and copy number of the chromosomally integrated reporter. The method used in this paper is inferior, since the PCR-based test used could fail by chance to detect a second reporter copy or, alternatively, concatemers of reporters integrated in tandem.

Reply: We thank the reviewer for this suggestion, as he/she correctly points out that splinkerette PCR, which is the method that we have used to study reporter integration, could in theory fail to detect additional integrations. As requested, we have now further determined the structure and copy number of the integrated DSB-Spectrum reporters using southern blotting, as suggested by the reviewer. This analysis, which was performed using two different restriction strategies and two different probes, clearly indicated that for each DSB-Spectrum clone a single copy of the full-length reporter construct was integrated. The southern blot results are now shown as figure S2B in the revised manuscript, and described in lines 170-184 on pages 6 and 7 of the revised manuscript.

Reviewer #3 (Remarks to the Author):

In the manuscript “Multi-Pathway DNA Double-Strand Break Repair Reporters Reveal Extensive Cross-Talk Between End-Joining, Single Strand Annealing, and Homologous Recombination” Kooij et al report the development of fluorescent Cas9-based reporter assay systems to monitor multiple DSB repair pathways simultaneously and to investigate crosstalk between them. The authors found that the inhibition of cNHEJ factors can increase the activity of SSA and HR and that the repair event by SSA is more common than generally thought. Importantly short-range resection by CtIP or Mre11 promoted both SSA and HR, while long-range resection by DNA2 or EXO1 promoted SSA and inhibited HR. This is the novel and key finding in this study. Currently the mechanism distinguishing SSA and HR is largely unknown. As outlined in more detail below, additional data is needed to support this finding. The assay systems developed here will be useful to investigate DSB repair pathway crosstalk and to understand outcomes following perturbation of the DSB signaling and repair network. Prior to publication, the following concerns should be addressed.

1. The observation that resection factors decide a DSB pathway, HR or SSA is the key finding of this paper. In the DSB-Spectrum_V3 cells, CtIP and Mre11 promoted SSA and HR. However, DNA2 and EXO1 promoted SSA but inhibited HR (Fig. 6). The authors should test if the similar effect can be detected in the SSA assay between Alu elements using the endogenous genomic loci shown in Fig 5, to confirm it in an additional context.

Reply: We thank the reviewer for acknowledging the significance of our finding that Exo1/DNA2 can determine SSA vs HR pathway choice. We have done the experiment suggested by the reviewer, and studied the role of Exo1 in the HR:SSA ratio at endogenous genomic loci. First, we have extended our analysis of SSA between Alu elements to two additional loci in the BTK and SPAST genes. Here we also find a clear contribution of Alu-mediated SSA of the Cas9-induced DSB, as is described in lines 499-517 on pages 16 and 17 of the revised manuscript, and as shown in figures 5A-G. Second, we have found that a DSB at the HBB gene, which is a target for gene editing therapeutics aimed at curing sickle cell disease, can be repaired by SSA between homologous regions in the HBB gene and the downstream HBD gene. This is described in lines 518-536 on page 17 of the revised manuscript, and shown in figures 5H, I.

Next, as suggested by the reviewer, we studied the impact of Exo1 depletion on SSA at the BTK, SPAST and HBB loci. At all three loci we see a clear reduction in SSA upon Exo1 knockdown, consistent with our results in DSB-Spectrum_V3 cells. To examine HR at the BTK

locus, we introduced an ectopic dsDNA repair template designed to knock-in a T7 primer binding site. HR efficiency could thus be monitored by PCR analysis. To study HR at the HBB locus, we performed sequence analysis to detect the HBB to HBD gene conversion product. These experiments confirmed that loss of Exo1 promoted HR at the HBB locus, validating our results with DSB-Spectrum. At the BTK locus, using an ectopic dsDNA repair template, however, we were surprised to find that loss of Exo1 did not affect HR frequency, suggesting that in this experimental setting Exo1 is neither required for HR nor inhibits it. We believe that the discrepant results between the BTK locus studies and the HBB/DSB-Spectrum studies likely reflect the difference between HR using the endogenous chromatinized sister chromatid (DSB-Spectrum/HBB) or a linear, non-chromatinized ectopic repair template (BTK). Nevertheless, in all cases studied we find that Exo1 is essential for SSA but not HR, and can even inhibit HR, such that loss of Exo1 will favor the HR:SSA ratio. These new data are now described in the revised manuscript in lines 567-614 at pages 19 and 20, and shown in figures 6E-L.

2. A-EJ (mutagenic EJ) was increased after the depletion of 53BP1 and DNA-PKcs in previous studies, Mol. Cell. 2016; 63:662–673 and NAR Cancer 2020; zcaa017, respectively. By contrast the authors report that mutagenic EJ was reduced by 53BP1si or NU7441 in Fig. 4. It is probably because mutagenic EJ detected by DSB-Spectrum_V2 and 3 was mediated by error prone cNHEJ. The authors should consider and discuss this possibility. BFP+ cells include samples repaired by cNHEJ. BFP- cells can be generated either by cNHEJ or by a-EJ in Fig. 2, 4, 5, 6.

Reply: Thank you for this comment. The reviewer correctly points out that loss of BFP in DSB-Spectrum_V2 and _V3 can be caused by either mutagenic c-NHEJ or by a-EJ. We apologize for not explaining this more clearly in our original manuscript. We have now corrected this in line 226 on page 8 of the revised manuscript. In addition, to specifically address the relative contributions of a-EJ vs error prone c-NHEJ to BFP loss, we have now performed a series of experiments using PolQ siRNAs and a DNA-PKcs inhibitor, including more detailed sequence analysis. These experiments clearly indicated that the majority of BFP loss is caused by mutagenic c-NHEJ, as suggested by the reviewer, and there is only a minor contribution of mutagenic repair by a-EJ. These experiments are described in lines 300-324 on pages 10-11 of the revised manuscript, and shown as new figure 3A and S5A.

3. The reporter sequence was stably integrated into human cell lines, but Cas9 and gRNA were transfected every time to monitor DSB repair. The transfection efficiency must influence the result outcome. How did the authors control for transfection efficiency in the experiments?

Reply: The construct that we transfect contains the Cas9 cDNA, the sgRNA, and a fluorescent protein encoding gene, in most cases iRFP(670). mCherry was used in initial experiments with DSB-Spectrum_V1 or _V2 cells. When cells were analyzed by flow cytometry, iRFP(670) expression was also measured as a proxy for transfection efficiency, and was additionally used to gate on transfected cells. We apologize for the lack of clarity on this procedure in our original manuscript, and we have now corrected this in the revised manuscript in lines 188-190 on page 7. We have also included a schematic of the experimental procedure as figure S3A.

4. +1 T insertion at CRISPR-induced DSBs was the most abundant among all Indel events (Fig. 3A). It has been reported that Cas9-catalyzed DNA cleavage produces 1 bp staggered ends rather than

blunt ends (Sci. Rep. 2016; 5:37584 & NAR Cancer 2020; zcaa017). This is a preferred substrate of c-NHEJ. This could explain the reason why NU7441 reduced the +1 T insertion.

Reply: We thank the reviewer for this prescient insight. The reviewer correctly points out that staggered cutting by Cas9 followed by c-NHEJ is the likely cause for the +1 bp insertion. We have now added this fact, as well as the corresponding reference (*Sci. Rep. 2016; 5:37584*) mentioned by the reviewer, to the revised manuscript (lines 261-262, page 9).

5. Knock down of endogenous protein level of POLQ and DNA2 after siRNA were not confirmed by immunoblotting in Sup Fig 1E and Fig 6B. Reduction of DNA2 mRNA was shown in Fig 6 C, such experiment was not performed for POLQ. Proper validation of knock-downs should be demonstrated by immunoblotting or qPCR.

Reply: We thank the reviewer for pointing this out, and apologize for omitting the validation of PolQ knockdown in our original manuscript. As we have not been able to find good-working commercially available antibodies recognizing human PolQ, we validated the knockdown using qPCR. We have included this validation as figures S3D (for DSB-Spectrum_V1) and S5B (DSB-Spectrum_V2).

6. Mutagenic EJ was detected with DSB-Spectrum _V2 and 3. However, the systems cannot distinguish the types of end-joining, cNHEJ and a-EJ. cNHEJ and a-EJ can both perform mutagenic EJ. Sequencing of DSB repair junctions is needed for further clarification. Indeed, TIDER analysis (sequencing analysis) identified products possibly repaired by a-EJ (Fig. 3B). The authors should reconsider the sentence in introduction “--- DSB-repair phenotypes that can easily be missed by commonly used sequencing approaches that analyze the DSB-repair junction, ”(Page 4, line 135-137)

Reply: The reviewer correctly notes that sequencing will distinguish between a-EJ and c-NHEJ, while DSB-Spectrum_V2 and _V3 cannot distinguish between those pathways. The sentence indicated by the reviewer was included to point out that large SSA deletions will be missed by sequencing of the repair junction, while DSB-Spectrum_V3 does detect those repair events. To avoid any confusion, we have now deleted this sentence from the introduction in the revised manuscript.

7. The authors should reconsider the sentence “As an alternative to c-NHEJ, DSBs can be repaired by Homologous Recombination (HR) (Page 3, Line 71-72)”. It is well established that the repair pathways are cell cycle dependent and affected by sequence of damage site. HR is a dominant pathway to repair DSBs especially in S/G2 cells.

Reply: Although not intended that way, we agree with the reviewer that this sentence might be misinterpreted as suggesting that HR functions as a back-up for c-NHEJ. To avoid any confusion, we have now changed the sentence to read: “A second DSB-repair pathway is Homologous Recombination (HR), which plays a particularly important role during the S/G2 phases of the cell cycle.” (lines 71-73, page 3 of the revised manuscript).

8. Page 17 Line 524, DBS should be DSB.

Reply: We have corrected this.

REVIEWERS' COMMENTS

Reviewer #1 (Remarks to the Author):

In this (revised) manuscript by van de Kooij and colleagues, the authors designed three different fluorescent CRISPR/Cas9-based double-strand break reporters, DSB-Spectrum_V1, DSB-Spectrum_V2, and DSB-Spectrum_V3. The V1 variant can distinguish error-free canonical non-homologous end-joining (c-NHEJ) from homologous recombination (HR,) V2 mutagenic- c-NHEJ repair versus HR, and V3 can distinguish c-NHEJ from HR, and single-strand annealing (SSA). After a thorough reading of the revised manuscript, I am pleased with the additional experiments performed. I do think that most of the comments are sufficiently addressed and have led to an improved manuscript. Moreover, I do think that additional experiments to assess SSA have made the conclusions more solid. Even though not all questions could be addressed via experiments, the explanations and additional figures led to a solid manuscript. Therefore, I would be supportive of publication in Nature Communications.

Reviewer #2 (Remarks to the Author):

The concerns raised previously have not been answered satisfactorily. First, the performance of the SSA reporter is suspect because the SSA product is not detected by conversion to a specific gain of wild type fluorescent marker gene. (The authors are directed to the SSA reporters developed by the Jasin and Stark labs as examples of the kind of positive fluorescent outcomes that are standard in the field). Second, as noted previously, depletion of BRCA1 has only a very modest effect on SSA frequencies in the experiments shown—much less dramatic than that observed previously. The authors requested a citation to back this comment up. Previous work has shown a substantial defect in SSA in BRCA1-depleted cells, with SSA reduced to ~20% of control values (e.g., data from the Stark lab, using an SSA reporter in the human osteosarcoma U2OS cell line PMID: 29212152). Technically, the FACS plots clearly show that the fluorescent populations often overlap the edges of the square gates used to define distinct repair outcomes, resulting in mutually overlapping FACS populations, the quantitation of which is bound to be inaccurate (e.g., Fig 2G, Fig 4C). This level of inaccuracy of measurement is problematic. Finally, the authors provide limited Southern blotting analysis of the reporter integrations in the cell lines they used. The puromycin resistance gene probe clearly shows multiple bands in at least one of the reporter cell lines studied. This finding raises the concern that some of the reporter cell lines studied might indeed contain multiple fragments of the reporter, with potential for additional sources of error in the attempt to quantify specific DSB repair outcomes.

Reviewer #3 (Remarks to the Author):

The authors have addressed all my questions and suggestions adequately. I do not have additional comments. I would recommend this interesting paper for publication in Nature Communications.

Reviewer response letter 2

Manuscript: NCOMMS-21-05354A

Title: Multi-Pathway DNA Double-Strand Break Repair Reporters Reveal Extensive Cross-Talk Between End-Joining, Single Strand Annealing, and Homologous Recombination

Authors: Bert van de Kooij, Alex Kruswick, Haico van Attikum, and Michael B. Yaffe

We would like to thank the reviewers for their careful evaluation of our revised manuscript. Please find our response below.

REVIEWER COMMENTS

Reviewer #1 (Remarks to the Author):

In this (revised) manuscript by van de Kooij and colleagues, the authors designed three different fluorescent CRISPR/Cas9-based double-strand break reporters, DSB-Spectrum_V1, DSB-Spectrum_V2, and DSB-Spectrum_V3. The V1 variant can distinguish error-free canonical non-homologous end-joining (c-NHEJ) from homologous recombination (HR,) V2 mutagenic- c-NHEJ repair versus HR, and V3 can distinguish c-NHEJ from HR, and single-strand annealing (SSA). After a thorough reading of the revised manuscript, I am pleased with the additional experiments performed. I do think that most of the comments are sufficiently addressed and have led to an improved manuscript. Moreover, I do think that additional experiments to assess SSA have made the conclusions more solid. Even though not all questions could be addressed via experiments, the explanations and additional figures led to a solid manuscript. Therefore, I would be supportive of publication in Nature Communications.

Reply: Thank you for this very positive evaluation of our revised manuscript.

Reviewer #2 (Remarks to the Author):

The concerns raised previously have not been answered satisfactorily. First, the performance of the SSA reporter is suspect because the SSA product is not detected by conversion to a specific gain of wild type fluorescent marker gene. (The authors are directed to the SSA reporters developed by the Jasin and Stark labs as examples of the kind of positive fluorescent outcomes that are standard in the field).

Reply: This comment is identical to comment 1 of the original report of this reviewer (reviewer #2; see original response letter for numbering of the reviewer comments). The reviewer basically states that any reporter other than those with a gain of fluorescence is flawed. We refer, again, to our original response letter in which we provided a detailed response that addressed this comment. In short:

- We performed knockdown experiments targeting a total of **eight** different DSB-repair proteins that would be expected to change the levels of SSA based on published literature

(Rad52, BRCA1, BRCA2, 53BP1, Mre11, CtIP, Exo1 and DNA2, shown in Fig. 4 and Fig. 6). In **ALL** cases the outcome was consistent with the BFP⁻,mCherry⁻ population being the result of SSA, which is a reduction in SSA-frequency upon knockdown of Rad52, BRCA1, Mre11, CtIP, Exo1 and DNA2, and an increase in SSA upon knockdown of 53BP1 and BRCA2.

- ***In addition, we directly demonstrated by PCR analysis and sequencing*** that DSB-repair of our reporter generates a very specific SSA-repair product. Importantly, we see a ~30% reduction of this specific SSA-repair product upon knockdown of Rad52 (Fig. 5C), which is ***exactly mirrored*** by the ~30% reduction of the BFP⁻,mCherry⁻ SSA-population detected by our reporter (Fig. 4E). Similarly, we see a ~2x increase of this SSA-repair product upon inhibition of DNA-PKcs (Fig. 5F), which is again ***exactly mirrored*** by a ~2x increase of the BFP⁻,mCherry⁻ SSA-population detected by our reporter (Fig. 4D). The most parsimonious explanation of these data and the knockdown experiments described above is that our SSA-reporter reliably detects changes in the levels of a specific DSB-repair product that is the consequence of SSA, thus validating DSB-Spectrum_V3 as an SSA-reporter system (in addition to it being a mut-EJ and HR reporter system).

Second, as noted previously, depletion of BRCA1 has only a very modest effect on SSA frequencies in the experiments shown—much less dramatic than that observed previously. The authors requested a citation to back this comment up. Previous work has shown a substantial defect in SSA in BRCA1-depleted cells, with SSA reduced to ~20% of control values (e.g., data from the Stark lab, using an SSA reporter in the human osteosarcoma U2OS cell line PMID: 29212152).

Reply: We appreciate that the reviewer added a citation showing strong reduction of SSA upon BRCA1 knockdown. As we already indicated in our original reply to this reviewer comment (comment #2 in the original report of this reviewer), there is large variation in the published literature with regards to the SSA phenotype upon BRCA1-loss. Whereas the citation mentioned by the reviewer observed a reduction to about 20% of control values, the citation that we discussed in our original response letter (PMID: 15485900) observed a reduction in SSA to about 55% of control values (on average, the error bars suggest large inter-experimental variation). Both citations used the SA-GFP reporter and were authored by Jeremy Stark, but differed with regards to the cell-line used (U2OS vs mES) and BRCA1 depletion method (siRNA vs KO). This clearly indicates that the effect of BRCA1-loss on SSA is context-dependent, and the fact that our experiments were done in a different cell-line using a different siRNA, with a reporter system integrated in a different site in the genome could therefore perfectly well explain the different effect-size that we observe.

In addition to overlooking differences in effect-size, we think it is incorrect to naively assume that one should expect the exact same degree of change in SSA when comparing two different reporter system with different DNA sequences and different integration sites. SA-GFP relies on a 280bp homology for SSA to occur while DSB Spectrum relies on 517bp of homology, possibly forming a more stable SSA intermediate that is likely more efficient at SSA. This is directly addressed by seminal work from James Haber (<https://journals.asm.org/doi/epdf/10.1128/mcb.12.2.563-575.1992>), showing a rather big SSA efficiency dependence on homology length, especially in the 200-500bp range, in yeast (Figure 6 specifically). Similarly, in human cells (U2OS) the SSA frequency strongly depends on homology length, as shown in figure 2B of this recent publication:

<https://journals.plos.org/plosgenetics/article?id=10.1371/journal.pgen.1008319>. In addition to homology length, repeat distance can also affect SSA efficiency, as shown by the lab of Jeremy Stark in mouse ES cells (<http://genesdev.cshlp.org/content/32/7-8/524.long>). Figure 1E of this paper indicates that decreasing the distance between the repeats enhances the SSA efficiency, in particular when the repeat distance drops below 3.3 kb. Thus, SA-GFP (repeat distance of 2.4 kb) might be more prone to SSA repair than DSB-Spectrum_V3 (repeat distance of 3.2 kb). Finally, the requirements for SSA repair factors are also dependent on the above mentioned parameters, as shown for example for Blm, which can either promote or suppress SSA depending on repeat distance and divergence (<https://doi.org/10.1016/j.celrep.2020.01.001>). The SA-GFP reporter that was referred to by the reviewer differs from DSB-Spectrum_V3 in repeat size (280 bp vs 507 bp) and repeat distance (~2.4 kb vs 3.2 kb). These reporter differences almost certainly account for any discrepant results between the reporter systems. However, there is no basis to suggest that one reporter is wrong and the other is right. In fact, the discrepancy most likely indicates, once again, context-dependency.

We would also like to refer to the original response letter to emphasize that we had already addressed this reviewer's comment experimentally by validating our BRCA1-phenotype using traditional PCR analysis of the SSA-repair product (see Reviewer Fig. 4 and reply to reviewer comment #2 in the original response letter).

As is always the case in science, one should not rely on a single assay/system/experimental readout to define a biological phenomenon. The fact that DSB-Spectrum is 100% in agreement with the reported dependencies of SSA on resection (CtIP/Mre11/Exo1/DNA2/BRCA1) as well as Rad52, the gold-standard SSA protein, should end the arguments put forth by the reviewer. If anything, the data gathered from DSB-Spectrum provide a much stronger claim for the scientific community that resection and Rad52 are mediators of bona-fide SSA, not solely SSA from the SA-GFP reporter construct, and in our mind further validates the utility of the SA-GFP reporter system that people have used for a while now.

Technically, the FACS plots clearly show that the fluorescent populations often overlap the edges of the square gates used to define distinct repair outcomes, resulting in mutually overlapping FACS populations, the quantitation of which is bound to be inaccurate (e.g., Fig 2G, Fig 4C). This level of inaccuracy of measurement is problematic.

Reply: This comment is identical comment #3 of the reviewer in the original response. We would again like to refer to the original response letter for a detailed reply, but will re-iterate the main points here in short. Loss of BFP and mCherry fluorescence requires degradation and cell division mediated dilution of the BFP/mCherry protein pool present at the time of gene editing. Thus, the separation between the BFP+ and BFP- population, as well as the mCherry+ and mCherry- population, will increase over time. Indeed, the data presented in supplementary figure 4A and 6B directly show that these populations separate well at 72h post reporter activation, and even better at 96h. Rather than waiting longer to achieve maximal separation, we chose to read out mostly at 72h, and gate tightly to include all BFP- or mCherry- events. This does result in a very minor population ending up in the BFP- or mCherry- gate also in cells with an uncut reporter (the AAVS1sg control cells). We simply correct for this when calculating the mut-EJ/SSA frequencies by subtracting the BFP- (or mCherry-) population in the AAVS1sg cells from the BFP- (or mCherry-) population in the BFPsg cells. That this

procedure results in accurate estimation of repair frequencies is validated by, among others, the observation that the changes in fluorescence-based analysis of SSA are identical to the changes in PCR-based analysis of SSA, as already explained above.

On a more general note, we do not believe that any reporter system gives an accurate quantification of the absolute levels of repair through a given pathway. We discuss this in detail in our recent review on reporter systems (PMID: 35237296). In short, reporter systems always assess repair of a very specific substrate integrated in a specific genomic location, rather than global genome repair, and the DSB in reporter systems is always generated by nucleases like Cas9 and I-SceI that keep cutting until the target site is sufficiently mutated to prevent further recognition. This applies to all reporters, including DR-GFP, SA-GFP and DSB-Spectrum. Furthermore, in gain-of-fluorescence reporters like SA-GFP a minority of SSA events might be missed because they do not result in perfect restoration of the GFP sequence, while in loss-of-fluorescence reporters like DSB-Spectrum_V3 (for the SSA and mut-EJ read-out) a small minority of events might be included that is actually not SSA/mut-EJ, but caused by other mutagenic processes resulting in the specific fluorescence changes. Nevertheless, reporters have been proven very useful, particularly for relative quantification, *i.e.* comparing repair frequencies between conditions. We expect DSB-Spectrum will be very useful for these kind of studies given its multi-pathway read-out.

Finally, the authors provide limited Southern blotting analysis of the reporter integrations in the cell lines they used. The puromycin resistance gene probe clearly shows multiple bands in at least one of the reporter cell lines studied. This finding raises the concern that some of the reporter cell lines studied might indeed contain multiple fragments of the reporter, with potential for additional sources of error in the attempt to quantify specific DSB repair outcomes.

Reply: The southern probe recognizing mCherry detects a single, very distinct band of the expected size in the lane containing DNA from HEK 293T DSB-Spectrum_V3 cells, both after SapI and after XbaI digestion of the genomic DNA (Fig. S2B, upper panels, lane 4). Thus, for DSB_Spectrum_V3, the southern analysis undisputedly validates single integration of the full-length reporter.

For the other DSB-Spectrum variants, which do not contain mCherry, we used a probe recognizing the puromycin resistance gene. This probe works well, although the results are slightly less 'clean' than with the mCherry probe, as the reviewer indicates. It recognizes DNA fragments of approximately 5.3 kb, 4.5 kb, and 2.8 kb in the SapI digested samples. These are non-specific background bands and not related to reporter fragments as evidenced by their appearance in the lane that contains DNA from a non-modified parental HEK 293T control cell-line (Fig. S2B, lower-left panel, lane 1). Lanes 2 and 3 on this blot contain DNA from HEK 293T DSB-Spectrum_V1 cells and HEK 293T DSB-Spectrum_V2 cells respectively. In both lanes only a single additional product is detected on top of these background bands, and each of these products has the exact size calculated based on single integration of the reporter (Fig. S2B, lower-left panel, lanes 2 and 3).

In the lanes containing XbaI digested DNA, the puro probe recognizes background products of about 11 kb and 3 kb (Fig. S2B, lower-right panel, lane 1). In addition to these background bands, a bright band is detected of the expected DBS-Spectrum-containing product, both in the HEK 293T DSB-Spectrum_V1 lane and the HEK 293T DSB-Spectrum_V2 lane (Fig. S2B, lower-right panel, lanes 2 and 3). In addition to this bright DSB-Spectrum band and the two non-specific background bands a fourth band is detected of 3.5 kb in lane 2 and 3.8 kb in lane 3 (Fig. S2B, lower-right panel, lanes 2 and 3). This could theoretically be an integrated reporter fragment containing (part of) the puromycin resistance gene. We consider this unlikely

Reviewer Figure

however, because it would mean that for both clones this fragment would have been missed upon SapI digestion. Also, the puro probe recognizes multiple bands in lane 4 containing DNA from DSB-Spectrum_V3 cells which do not contain a puromycin resistance gene. This indicates that low-intensity non-specific bands can appear when southern blotting with this probe. Lastly, in earlier southern analyses we had also used a southern probe recognizing the PGK promoter upstream of the puro (in V1 and V2) or mCherry (in V3) gene (see Reviewer Figure). We had not included this analysis in the original manuscript because the probe overall generated low intensity bands, leading us to switch to the mCherry and puro probes. Nevertheless, only a single specific product of the length predicted based on its integration site was detected for all clones (see Reviewer Figure). Taken together therefore, our extensive southern analysis using three different probes strongly suggests that only one full-length reporter is integrated in each clone.

Noteworthy, on top of the southern analysis we have also performed splinkerette PCR analysis to determine reporter copy number and integration (Fig. S2B). For all clones we retrieved only a single reporter integration site. Also, the lentiviral infection to generate the clones was performed at very low MOI (infection rate of lower than 5%), making the chance of getting multiple integrations extremely low. Added to the southern analysis, this makes multiple integrations extremely unlikely.

Reviewer #3 (Remarks to the Author):

The authors have addressed all my questions and suggestions adequately. I do not have additional comments. I would recommend this interesting paper for publication in Nature Communications.

Reply: Thank you for your very positive evaluation of our revised manuscript.